# T follicular helper and T follicular regulatory cells have different TCR specificity

Ana Raquel Maceiras[1,2], Silvia Cristina Paiva Almeida[1,2], Encarnita Mariotti-Ferrandiz[3,4,5], Wahiba Chaara[3,4,5], Fadi Jebbawi[4], Adrien Six[3,4,5], Shohei Hori[6], David Klatzmann[3,4,5], Jose Faro[2,7,8,9,*] & Luis Graca[1,2,*]

Immunization leads to the formation of germinal centres (GCs) that contain both T follicular helper (Tfh) and T follicular regulatory (Tfr) cells. Whether T-cell receptor (TCR) specificity defines the differential functions of Tfh and Tfr cells is unclear. Here we show that antigen-specific T cells after immunization are preferentially recruited to the GC to become Tfh cells, but not Tfr cells. Tfh cells, but not Tfr cells, also proliferate efficiently on restimulation with the same immunizing antigen *in vitro*. *Ex vivo* TCR repertoire analysis shows that immunization induces oligoclonal expansion of Tfh cells. By contrast, the Tfr pool has a TCR repertoire that more closely resembles that of regulatory T (Treg) cells. Our data thus indicate that the GC Tfh and Tfr pools are generated from distinct TCR repertoires, with Tfh cells expressing antigen-responsive TCRs to promote antibody responses, and Tfr cells expressing potentially autoreactive TCRs to suppress autoimmunity.

[1] Instituto de Medicina Molecular, Faculdade de Medicina, Universidade de Lisboa, 1649-028 Lisboa, Portugal. [2] Instituto Gulbenkian de Ciência, 2780-156 Oeiras, Portugal. [3] Sorbonne Universités, UPMC Univ Paris 06, UMRS 959, Immunology-Immunopathology-Immunotherapy (i3), F-75005 Paris, France. [4] INSERM, UMRS 959, Immunology-Immunopathology-Immunotherapy (i3), F-75005 Paris, France. [5] AP-HP, Hôpital Pitié-Salpêtrière, Biotherapy and Département Hospitalo-Universitaire Inflammation-Immunopathology-Biotherapy (i2B), F-75651 Paris, France. [6] Laboratory for Immune Homeostasis, RIKEN Center for Integrative Medical Sciences, Kanagawa 230-0045, Japan. [7] Área de Immunoloxía, Facultade de Bioloxía, Universidade de Vigo, 36310 Vigo, Spain. [8] Centro de Investigacións Biomédicas (CINBIO), Universidade de Vigo, 36310 Vigo, Spain. [9] Instituto Biomédico de Vigo, 36310 Vigo, Spain. * These authors jointly supervised this work. Correspondence and requests for materials should be addressed to L.G. (email: lgraca@medicina.ulisboa.pt).

Thymus-dependent humoral immune responses are not only critical for protection against pathogens but are also a central protective mechanism of most vaccines. These antibody-mediated responses depend on germinal centres (GCs)—anatomical structures inside the B-cell zone—where T follicular helper (Tfh) cells interact with and provide help to B cells, enabling affinity maturation and isotype switching[1]. Affinity maturation is a critical event in the GC reaction in which B cells edit their B-cell receptor (BCR) and undergo a selection process leading to higher receptor affinity. However, during affinity maturation, autoreactive BCRs may be generated, resulting in production of autoantibodies and the potential for autoimmune disease. Several autoimmune diseases are characterized by formation of ectopic GCs and production of autoantibodies[2].

Tfh cells are required for GC formation and maintenance[3–7], and Foxp3$^+$ T follicular regulatory (Tfr) cells participate in the regulation of GC reactions[8–12]. Lack of Tfr cells or an altered Tfr:Tfh ratio can increase the risk of autoimmunity and autoantibody production[13–16]. This contribution of Tfr cells to the prevention of autoimmunity has been detected in several experimental models of autoimmunity and inferred from human pathology[13–17].

Here we test the hypothesis that populations of Tfh and Tfr cells have different T-cell receptor (TCR) repertoires, leading to different antigenic targets for effector versus regulatory action. Protective immune responses are promoted by Tfh cells, which, with a TCR repertoire specific for an immunizing antigen, provide help to B cells and enable BCR affinity maturation, whereas the Tfr cell TCR repertoire, which is predominantly autoreactive, enables these cells to suppress autoreactive affinity-matured B-cell clones, thus preventing autoantibody-mediated autoimmunity.

Using antigen-specific CD4$^+$ T cells from TCR-transgenic mice, we demonstrate that recruitment of Tfh cells into GCs is predominantly controlled by specificity for the immunizing antigen. By contrast, recruitment of Tfr cells for the same GCs was not biased towards specificity for the immunizing antigen. These findings are confirmed in wild-type (WT) mice using major histocompatibility complex (MHC) class II tetramers: while we detect a large population of tetramer-positive Tfh cells, almost no tetramer-positive Tfr cells are found. In addition, we use an independent approach, analysing the TCR diversity from sorted T-cell subsets (including Tfh and Tfr) to demonstrate that Tfh cells from GCs induced by immunization with a defined antigen present oligoclonal expansions that are not observed on the Tfr subset. Moreover, the Tfr cell TCR repertoire closely resembles the thymic regulatory T (Treg) cell repertoire. Thus, our data not only confirm that Tfh cells differentiate predominantly from naive Foxp3$^-$ T cells and that Tfr cells originate from thymic Foxp3$^+$ Treg cells but also show that the ontogeny of Tfh and Tfr cells corresponds to a distinct TCR usage.

## Results

**Tfr cells differentiate from thymic Foxp3$^+$ Treg cells.** We had previously shown that under lymphopenic conditions, immunization with a foreign antigen leads to GC formation containing Tfr cells that differentiate from adoptively transferred thymic Foxp3$^+$ Treg cells[8]. To exclude a potential artefact elicited from lymphopenic conditions we now investigated, using congenic markers, the precursors of Tfr cells following immunization in two distinct genetic backgrounds (Fig. 1). Magnetic-activated cell sorting (MACS)-purified OVA-specific TCR-transgenic CD4$^+$ T cells from OT-II.$Rag^{-/-}$ or DO11.10.$Rag^{-/-}$ mice, devoid of thymic Foxp3$^+$ Treg cells, were adoptively transferred into naive

C57BL/6 or Balb/c hosts, respectively (Fig. 1a,b). Recipient mice were subsequently immunized with OVA in incomplete Freund's adjuvant (OVA-IFA) in the footpad and draining popliteal lymph nodes (LNs) were analysed by flow cytometry, at the peak of GC response, when higher numbers of Tfh and Tfr cells can be obtained (day 11)[8,18]. Popliteal LNs were found to contain Tfh populations derived to a great extent from transferred TCR-transgenic cells, while Tfr cells derived exclusively from endogenous T cells (Fig. 1c,d). On the contrary, TCR-transgenic CD4$^+$ T cells from OT-II.$Rag^+$ mice that, unlike $Rag$-deficient counterparts, harbour a population of TCR-transgenic thymic Foxp3$^+$ Treg cells (Fig. 1e), can support the differentiation of Tfr cells, following adoptive transfer into T-cell-deficient mice (Fig. 1f). Indeed, when OT-II.$Rag^+$ cells were transferred into $TCR\beta^{-/-}$ mice we found that they readily gave rise to a population of Tfr cells. Note that the popliteal LN allows the study of GCs driven by the immunizing antigen since Tfh and Tfr cells, abundant in immunized mice, are virtually absent on equivalent LNs of non-immunized C57BL/6 mice (Fig. 1g,h). These results show that while adoptive transfer of thymic-derived Foxp3$^+$ Treg cells can differentiate into Tfr cells Foxp3$^-$ T cells only differentiate into Tfh.

**No preferential recruitment of antigen-specific Tfr cells.** To assess the specificity requirements of Tfr cells, we took advantage of OT-II.$Rag^+$ mice. OVA-specific TCR-transgenic CD4$^+$ T cells from these mice were transferred into congenic C57BL/6 mice subsequently immunized with OVA-IFA or a control antigen (β-lactoglobulin, βLG; Fig. 2a). Again, using congenic markers, it was possible to quantify the frequency of TCR-transgenic cells within the Tfh and Tfr populations (Fig. 2a,b). We found that a large fraction of Tfh cells in mice immunized with OVA were derived from OVA-specific precursors, while those TCR-transgenic precursors were almost excluded from Tfr cells even in OVA-immunized mice (Fig. 2b,c). To exclude that the low number of TCR-transgenic Foxp3$^+$ Tfr cells observed was due to competition by a much larger number of endogenous cells, we performed adoptive cell transfers into T-cell-deficient mice, allowing competition of the same number of TCR-transgenic and WT T cells. Thus, the same number of CD4$^+$ T cells from OT-II.$Rag^+$ and WT mice was co-transferred into $TCR\beta^{-/-}$ mice followed by immunization with OVA or a control antigen (Fig. 2d). Under those conditions, and 11 days after immunization, TCR-transgenic cells became vastly over-represented among Tfh cells in mice immunized with OVA (Fig. 2e). In addition, we found that TCR-transgenic cells were able to differentiate into Tfr, but their frequency ($\sim 1/5$ of Tfr cells) did not change significantly regardless of the immunizing antigen. In addition, the percentage of OT-II.$Rag^+$ Treg cells that differentiated into Tfr is $\sim 8\%$ in both immunizations although, as expected, in OVA-immunized mice the percentage of OT-II.$Rag^+$ conventional T (Tconv) cells that differentiate into Tfh is higher than in βLG-immunized mice (Supplementary Fig. 1).

Foxp3$^+$ Treg cells from OT-II.$Rag^+$ mice co-express endogenous TCR chains, in addition to the transgene, that allow their thymic selection as Treg cells[19]. In fact, WT T cells can also express more than one TCR, due to recombination of both TCRα chains[20]. We found that, as previously described, a proportion of Treg cells ($\sim 30\%$) do not co-express the transgenic TCR chains Vα2 and Vβ5 unlike Tconv cells, which are virtually all double-positive. Nevertheless, on adoptive transfer into $TCR\beta^{-/-}$ mice and immunization, the percentage of OT-II.$Rag^+$ Treg and Tfr cells co-expressing Vα2 and Vβ5 remained unchanged (Fig. 2f). Therefore, we could not find a preferential enrichment of Vα2

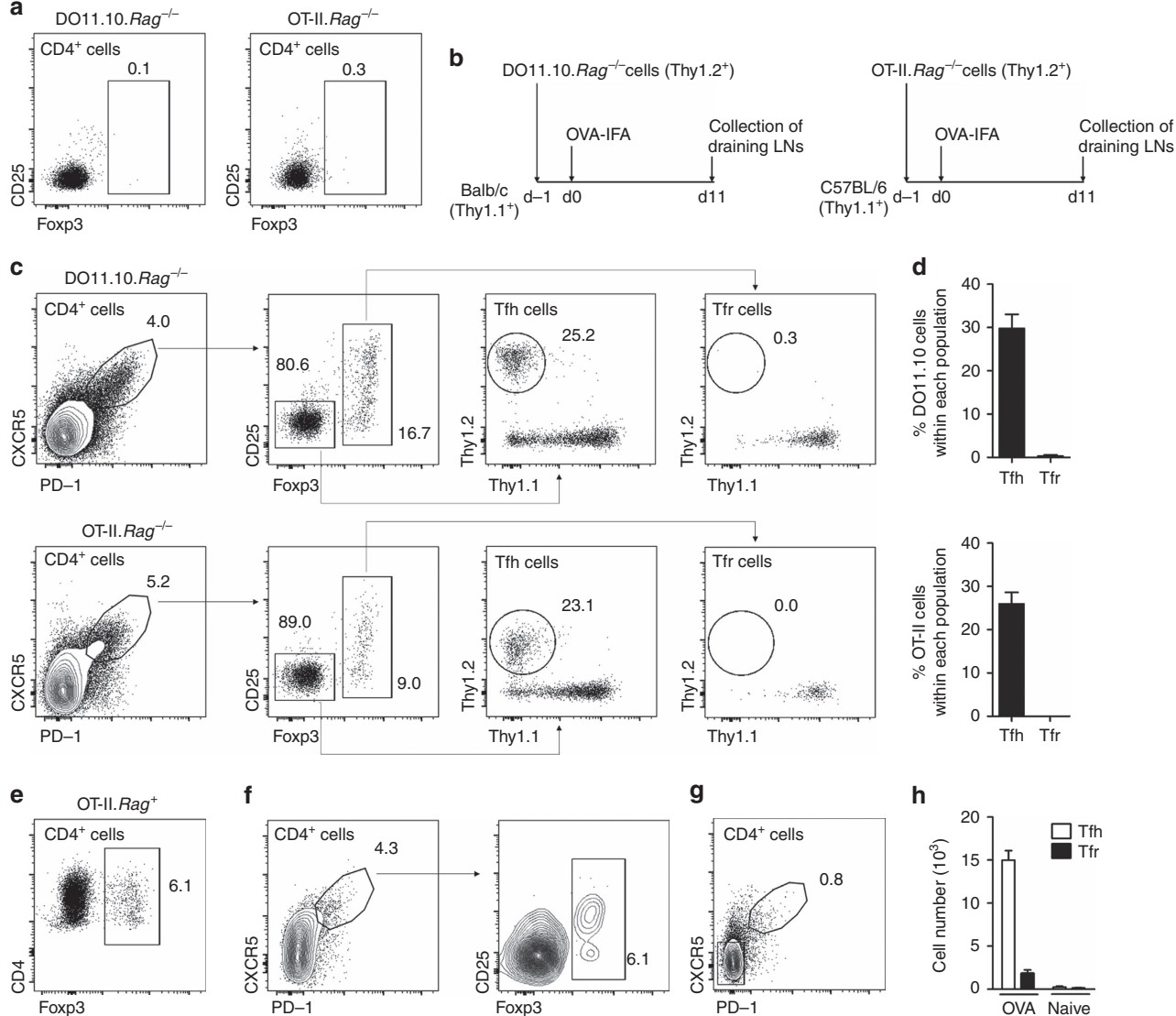

**Figure 1 | Tfr cells do not differentiate from Foxp3$^-$ T-cell precursors.** (**a**) CD4$^+$ T cells from OVA-specific OT-II.$Rag^{-/-}$ or DO11.10.$Rag^{-/-}$ mice are devoid of Foxp3$^+$ Treg cells. (**b**) 10$^6$ CD4$^+$ T cells from OT-II.$Rag^{-/-}$ or DO11.10.$Rag^{-/-}$ mice were adoptively transferred into, respectively, C57BL/6 or Balb/c hosts subsequently immunized with OVA-IFA in the footpad. At day 11 popliteal LNs were analysed by flow cytometry. (**c**) Gating strategy for detection of DO11.10.$Rag^{-/-}$ (upper panel) or OT-II.$Rag^{-/-}$ (bottom panel) within Tfh and Tfr cell populations. (**d**) While Tfh (CD4$^+$CXCR5$^+$PD-1$^+$Foxp3$^-$) cells contained ∼25–30% TCR-transgenic cells, those adoptively transferred cells could not be detected among the Tfr (CD4$^+$CXCR5$^+$PD-1$^+$Foxp3$^+$) population in any of the two genetic backgrounds. Mean ± s.e.m. are presented for $n = 5$. (**e**) OT-II.$Rag^+$ mice have Foxp3$^+$ Treg cells. (**f**) Adoptive transfer of CD4$^+$ T cells from OT-II.$Rag^+$ mice into $TCR\beta^{-/-}$ hosts followed by immunization as described in **b**. Under these conditions the transferred TCR-transgenic cells originated both Tfh and Tfr cells. (**g**) Relative frequency of T follicular cells in popliteal LNs of non-immunized C57BL/6 mice. (**h**) Absolute number of Tfh and Tfr cells within popliteal LNs from non-immunized C57BL/6 mice compared to OVA-immunized mice. Mean ± s.e.m. is presented for $n = 3$.

Vβ5 double-positive cells, more likely to be specific for the immunizing antigen, within the regulatory populations after OVA immunization.

As the above observations may question the ability of Foxp3$^+$ cells from OT-II.$Rag^+$ mice to properly respond to OVA stimulation, we confirmed that OT-II Treg cells can be activated by OVA. We cultured sorted Treg (CD4$^+$CD25$^+$GITR$^+$) cells from OT-II.$Rag^+$ and C57BL/6 WT mice in presence of interleukin-2 (IL-2) and OVA-pulsed bone marrow-derived dendritic cells (DCs; Fig. 2g,h and Supplementary Fig. 2a). We found that OVA-loaded DCs, unlike DCs loaded with a control antigen, promoted proliferation of OT-II Treg cells to similar extent as stimulation with anti-CD3. Moreover, WT Treg cells show lower proliferation and cell numbers when cultured in the

presence of both antigens compared to cultures with anti-CD3. Of note, it was previously described that some unspecific proliferation is induced on Treg cells when cultured with activated DCs in the presence of exogenous IL-2 (refs 21,22), which is in line with what we observed in the cultures of WT Treg cells with OVA/βLG-loaded DCs and OT-II Treg cells with βLG-loaded DCs. These data demonstrate that OT-II Treg cells, in spite of the co-expression of endogenous TCR chains, specifically recognize and proliferate in presence of OVA stimulation. As a consequence, the poor differentiation of OT-II Treg cells into Tfr cells following OVA immunization does not appear to be due to loss of OVA reactivity.

The low recruitment of OT-II.$Rag^+$ Treg cells as Tfr cells could be due to an intrinsic characteristic of OT-II transgenic

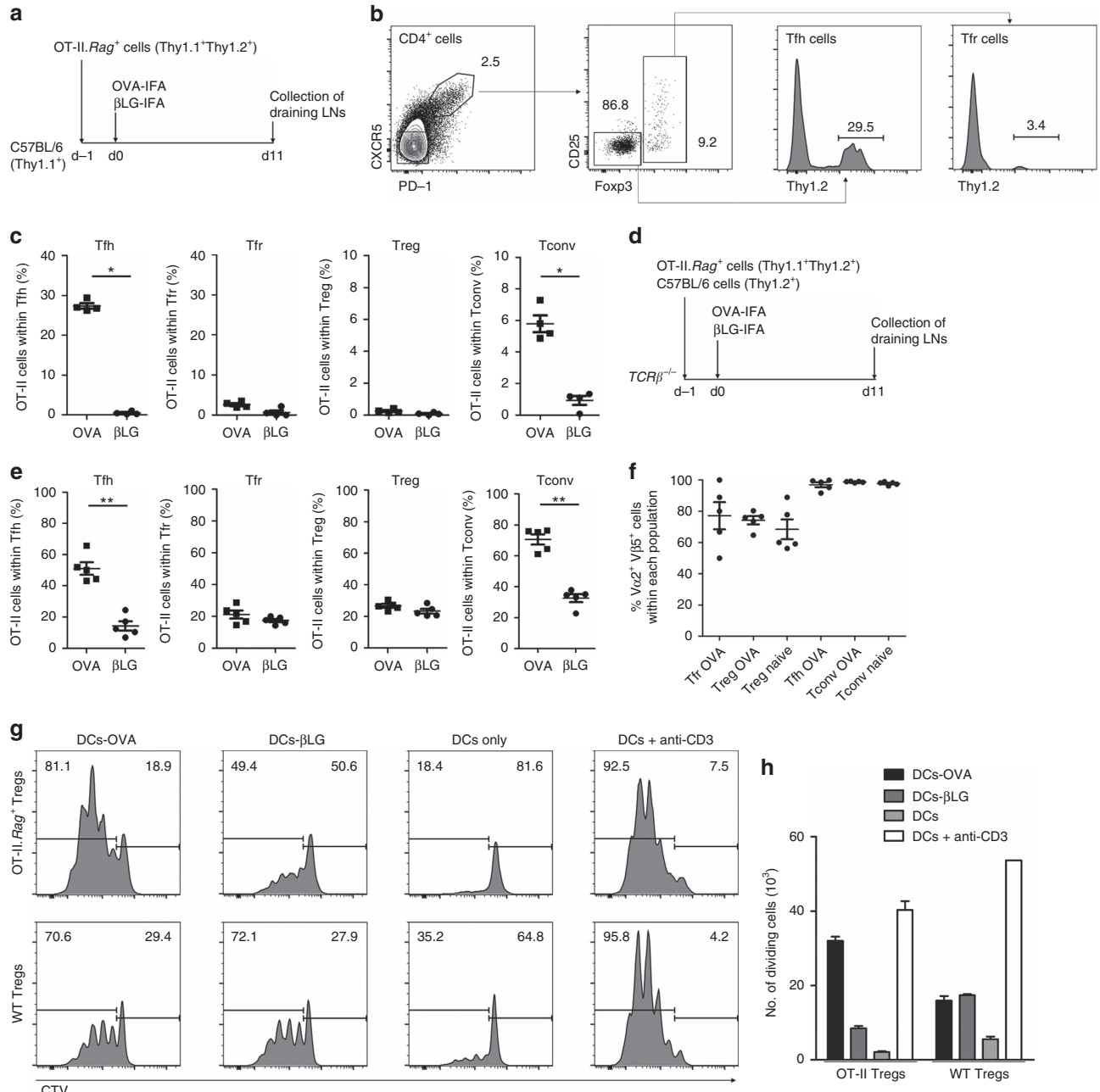

**Figure 2 | No preferential accumulation of OVA-specific cells within the Tfr population.** (**a**) $10^7$ CD4$^+$ T cells from OT-II.*Rag*$^+$ mice were transferred into C57BL/6 hosts, subsequently immunized with OVA-IFA or βLG-IFA in the footpad. (**b**) Popliteal LNs were analysed for the presence of OT-II.*Rag*$^+$ TCR-transgenic cells within Tfh and Tfr populations based on Thy1.2 staining. Tfh cells were defined as CD4$^+$CXCR5$^+$PD-1$^+$Foxp3$^-$ and Tfr cells as CD4$^+$CXCR5$^+$PD-1$^+$Foxp3$^+$. The same gating strategy was applied on Treg (CD4$^+$CXCR5$^-$PD-1$^-$Foxp3$^+$) and Tconv (CD4$^+$CXCR5$^-$PD-1$^-$Foxp3$^-$) cells to determine the percentage of Thy1.2$^+$ OT-II.*Rag*$^+$ TCR-transgenic T cells within those two populations. (**c**) OVA-specific TCR-transgenic cells were over-represented within Tfh and Tconv populations in mice immunized with OVA (*$P < 0.05$ using two-tailed non-parametric Mann–Whitney *U*-tests). Within Tfr and Treg cells there was no significant increase in OVA-specific cells in mice immunized with OVA, compared to βLG-immunized animals. (**d**) An equal number ($10^7$) of CD4$^+$ T cells from OT-II.*Rag*$^+$ and C57BL/6 mice were transferred into T-cell-deficient *TCRβ*$^{-/-}$ mice, subsequently immunized as described above. (**e**) Under these conditions there was an even greater representation of OVA-specific cells within Tfh and Tconv populations (*$P < 0.05$ and **$P < 0.01$ using two-tailed non-parametric Mann–Whitney *U*-tests). Although ~20% of Tfr cells derived from the OVA-specific TCR-transgenic population, that frequency remained similar in mice immunized with OVA or βLG. Similar results were obtained in two additional independent experiments, all with $n = 5$. (**f**) Frequency of Vα2 Vβ5 double-positive cells within Tfr, Treg, Tfh and Tconv populations from naive or OVA-immunized mice. (**g**) Fluorescence-activated cell sorting-purified C57BL/6 and OT-II.*Rag*$^+$ Treg (CD4$^+$CD25$^+$GITR$^+$) cells were labelled with CellTrace Violet (CTV) and cultured for 3 days in presence of IL-2 and bone marrow DCs loaded with OVA or βLG. In control groups T cells were cultured with unloaded DCs with or without soluble anti-CD3. Histograms are representative of Treg (CD4$^+$TCRβ$^+$CD25$^+$Foxp3$^+$) cell proliferation at the end of the culture. (**h**) Quantification of the number of proliferating cells. Culture triplicates are presented on the histogram and are representative of three independent experiments. Mean ± s.e.m. are presented in all graphs.

cells. To test this possibility, we repeated the experiment with a second TCR-transgenic model. We transferred, simultaneously, an equal number of CD4$^+$ T cells from OT-II.$Rag^+$ and P25.$Rag^+$ mice (TCR-transgenic specific for the residues 280–294 of *Mycobacterium tuberculosis* Ag85B) that also have thymic-derived Treg cells (Fig. 3a,b)[23]. Recipient mice were immunized in the footpad with OVA$_{323–339}$ or Ag85B$_{280–294}$ peptides coupled to bovine serum albumin (BSA) in IFA. We used peptides coupled to a carrier protein (BSA) because it has been reported that immunization with linear epitopes may fail to generate optimal antibody responses[24]. We found that Tfh cells from the draining LNs have a substantial frequency increase of TCR-transgenic cells specific for the immunizing antigen (a high frequency of OT-II cells in mice immunized with OVA$_{323–339}$BSA, and P25 cells in mice immunized with Ag85B$_{280–294}$BSA; Fig. 3c,d). Within the Tfr population there was only a small increase of TCR-transgenic cells specific for the immunizing antigen.

These results suggest that while TCR-transgenic cells specific for the immunizing antigen are able to become Tfr cells, those cells are not preferentially selected into the Tfr pool.

**Tfr cells do not recognize the immunizing antigen.** To obtain an independent validation of our findings without the use of TCR-transgenic mice, we used two MHC-II I-A$^b$ tetramers: a phycoerythrin (PE)-labelled tetramer containing an OVA peptide sequence (AAHAEINEA) to identify OVA-specific T cells and an allophycocyanin (APC)-labelled tetramer containing the Ag85B peptide sequence (FQDAYNAAGGHNAVF) to identify Ag85B-specific T cells. Antigen-specific tetramer$^+$ cells were detected on draining LNs of C57BL/6 mice 11 days after immunization with several combinations of antigens and adjuvants (Fig. 4a). In all immunizations with OVA$_{323–339}$, the number of OVA-tetramer$^+$ cells was increased among the Tfh population compared to mice that were not immunized with this

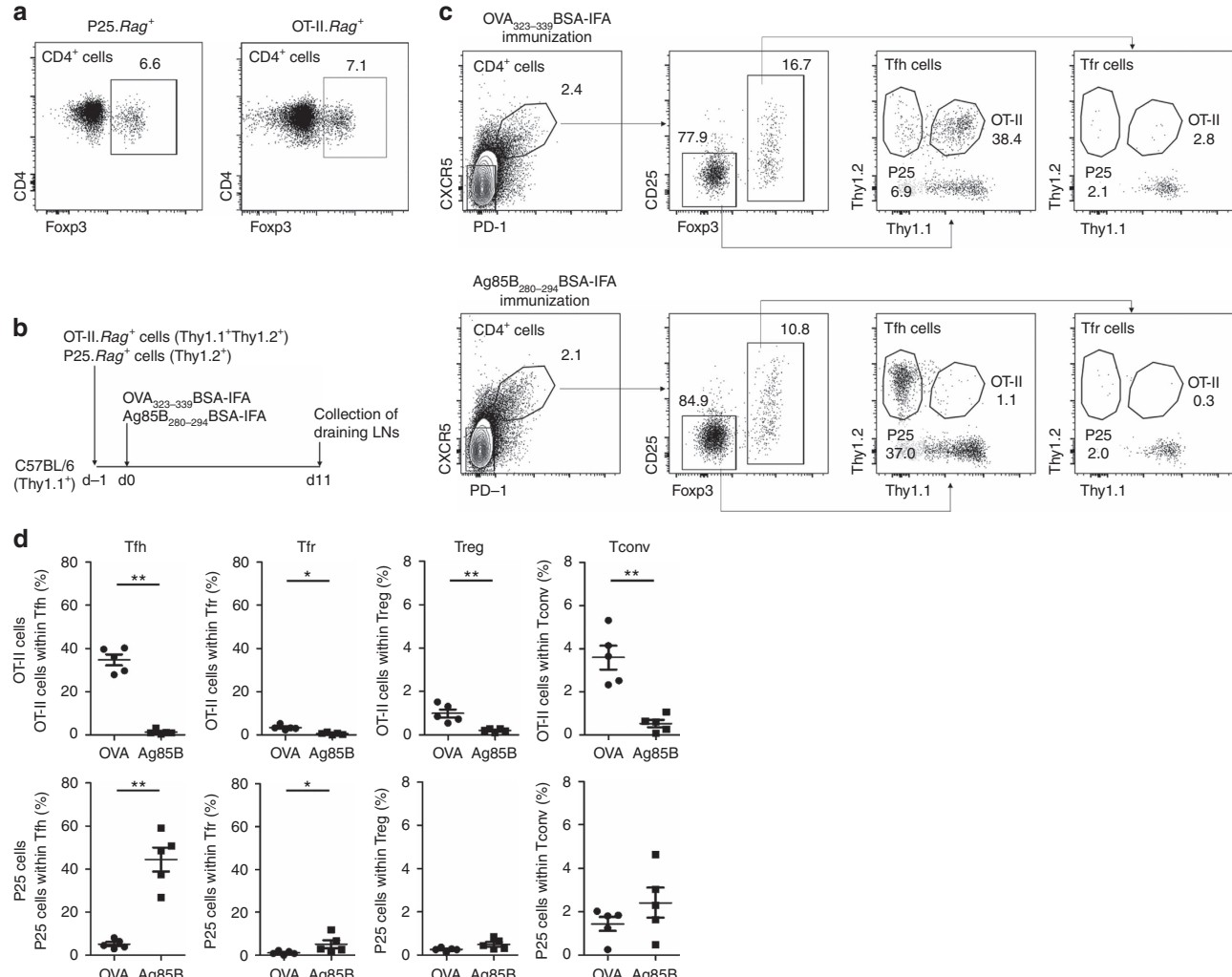

**Figure 3 | P25 cells are not preferentially recruited into the Tfr pool.** (**a**) P25.$Rag^+$ mice have similar frequency of thymic Treg cells (6.0 ± 0.9%) as OT-II.$Rag^+$ mice (6.4 ± 0.7%). (**b**) C57BL/6 mice were transferred simultaneously with 10$^7$ CD4$^+$ T cells from OT-II.$Rag^+$ and P25.$Rag^+$ mice, and subsequently immunized with either OVA$_{323–339}$BSA-IFA or Ag85B$_{280–294}$BSA-IFA in the footpad. (**c**) Gating strategy to determine the percentage of OT-II.$Rag^+$ and P25.$Rag^+$ cells within Tfh and Tfr populations in mice immunized with OVA$_{323–339}$BSA-IFA (upper panel) or Ag85B$_{280–294}$BSA-IFA (bottom panel). (**d**) T-cell subsets from draining LNs show that mice immunized with OVA$_{323–339}$ have a large accumulation of OVA-specific cells within the Tfh and Tconv populations while, conversely, Ag85B$_{280–294}$-immunized mice accumulate P25-specific T cells among Tfh cells (*$P < 0.05$ and **$P < 0.01$ using two-tailed non-parametric Mann–Whitney $U$-tests). We observed a very small increase of T cells specific for the immunizing antigen among the Tfr population. In all graphs, mean ± s.e.m. are presented. Data are representative of three independent experiments, each with $n = 5$.

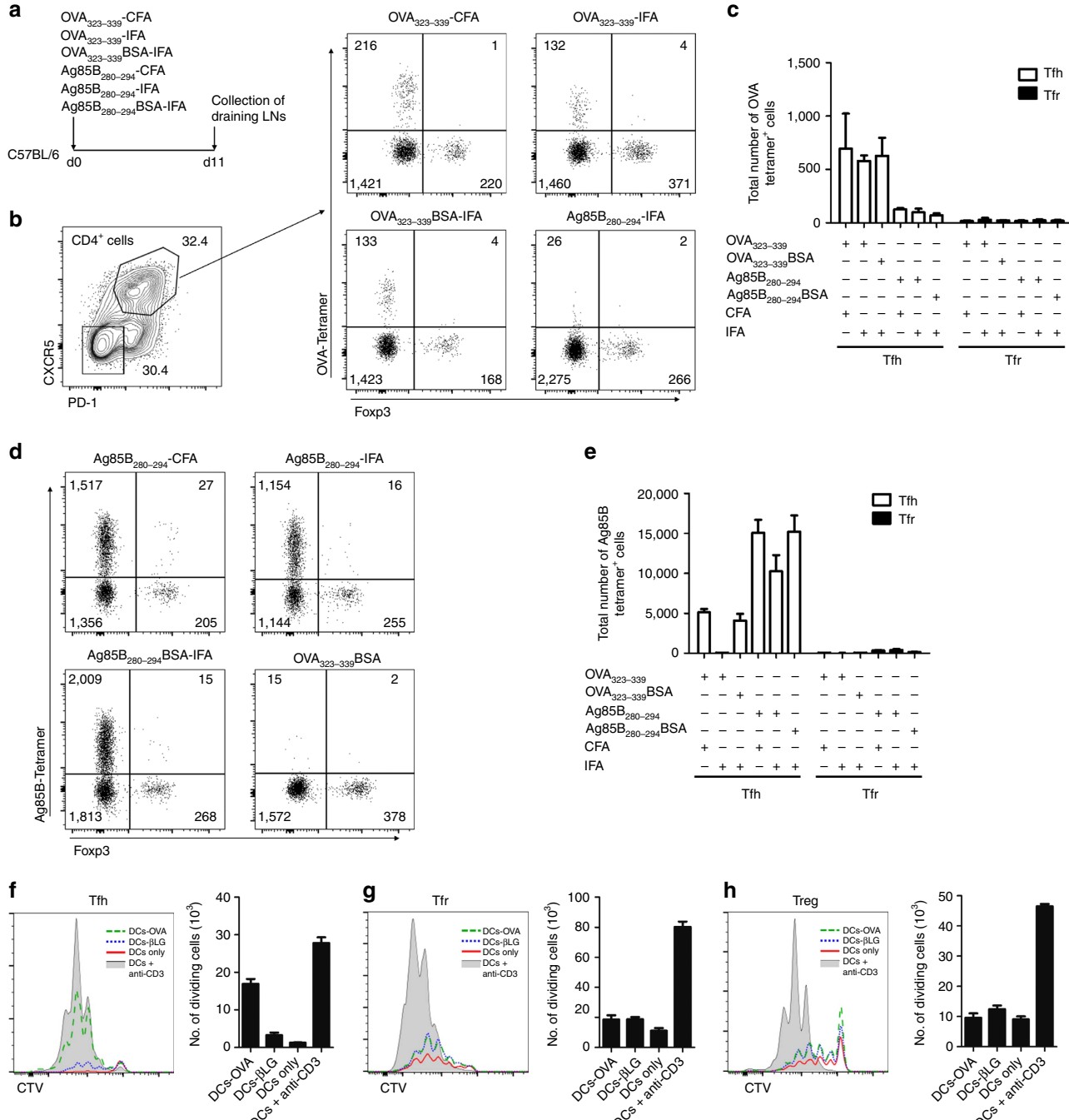

**Figure 4 | Tfr cells neither bind antigen-specific tetramers nor proliferate *in vitro* on restimulation.** (**a**) C57BL/6 mice were immunized with four different antigens combined with two different adjuvants on a total of six different immunizations. On day 11, draining LNs were collected for tetramer-binding cell detection. (**b**) Gating strategy to identify OVA-tetramer$^+$ Tfh and Tfr cells within CD4$^+$ T cells. Relative percentages for the two gates are presented on the contour plot on the left, while event counts for each quadrant are presented on the four scatter plots on the right. The Tfh population from mice immunized with the three conditions containing OVA$_{323-339}$ was enriched on cells with TCRs capable of binding the OVA-tetramer. Such OVA-tetramer$^+$ cells were almost absent in the Tfr population in all immunizations. Scatter plots are representative of the results obtained for each of the immunizations with OVA$_{323-339}$ and of the controls immunized with different formulations of Ag85B$_{280-294}$. (**c**) Total number of OVA-tetramer$^+$ cells in the draining LNs. (**d**) Scatter plots of Ag85B-tetramer$^+$ Tfh and Tfr cells within follicular CD4$^+$ T cells; the numbers represent the event counts for each quadrant. As observed for the OVA-tetramer, a large population of tetramer$^+$ Tfh cells was observed on immunizations containing Ag85B$_{280-294}$ peptide, while low numbers of tetramer$^+$ Tfr cells were found in all immunizations. Scatter plots are representative of the results obtained for each of the immunizations with Ag85B$_{280-294}$ and of the controls immunized with OVA$_{323-339}$. (**e**) Total number of Ag85B-tetramer$^+$ cells in the draining LNs. Mean ± s.e.m. are presented. Data are representative of two independent experiments, each with $n = 5$. (**f–h**) Histograms of proliferation and bar graphs of total cell numbers of sorted Tfh (CD4$^+$CXCR5$^+$PD-1$^+$Foxp3$^-$) (**f**), Tfr (CD4$^+$CXCR5$^+$PD-1$^+$Foxp3$^+$) (**g**) and Treg (CD4$^+$CXCR5$^-$PD-1$^-$Foxp3$^+$) cells (**h**) from OVA-immunized *Foxp3$^{hCD2}$* mice cultured for 3 days with DCs loaded with OVA or βLG or unloaded DCs with or without anti-CD3 and in the presence of IL-2. Only Tfh cells show higher proliferation and total numbers on OVA-pulsed DCs cultures compared to βLG-pulsed ones. Two independent experiments were performed: Mean ± s.e.m. of one experiment triplicates are presented on the histograms.

peptide (Fig. 4b,c). The same pattern was observed in mice immunized with $Ag85B_{280-294}$, where we found larger populations of $Ag85B$-tetramer$^+$ Tfh cells (Fig. 4d,e). However, in neither case did we observe an enrichment of tetramer$^+$ cells within the Tfr population (Fig. 4b–e). Once again, these results demonstrate that Tfr cells, unlike Tfh cells, do not preferentially contain cells specific for the non-self antigen driving the GC response.

To investigate whether Tfr and Tfh TCRs recognize different peptides from the same antigen, we assessed proliferation of sorted Tfh ($CD4^+CXCR5^+PD-1^+Foxp3^-$), Tfr ($CD4^+CXCR5^+PD-1^+Foxp3^+$) and Treg ($CD4^+CXCR5^-PD-1^-Foxp3^+$) cells stimulated with DCs loaded with the immunizing antigen to verify whether they specifically proliferate with these signals (Supplementary Fig. 2b). Tfh cells from OVA-immunized C57BL/6 Foxp3 reporter mice ($Foxp3^{hCD2}$)[25] were cultured with DCs loaded with OVA or βLG as a control antigen. Tfh cells showed higher proliferation and survival on OVA cultures compared to DCs loaded with βLG or unloaded DCs, demonstrating that this population specifically recognizes antigen signals (Fig. 4f). On the contrary, Tfr cells cultured with OVA presented some proliferation (probably due to the same nonspecific effect observed on Treg cell cultures on Fig. 2g,h) but that proliferation was not higher than in the presence of βLG or unloaded DCs (Fig. 4g). The presence of anti-CD3 induced a higher proliferation rate and survival of Tfr cells that translated into higher number of cells at the end of culture compared to the other conditions. Therefore, if Tfr cells were specific for the immunizing antigen, TCR signalling would have led to higher proliferation and cell numbers in the end of culture (Fig. 4g). Of note, follicular T cells seem to be fragile, with low survival capacity when they are not receiving TCR signalling. In fact, non-proliferating cells die quickly in culture and only cells that undergo some degree of background proliferation survive. Tfh, but not Tfr, can be rescued when the immunizing antigen is added to the culture. As Tfr cells, Treg cells sorted from immunized mice also show lower proliferation and cell numbers when cultured with DCs loaded with OVA or βLG compared to the culture in the presence of anti-CD3 (Fig. 4h) and as previously observed on Fig. 2g,h. Taken together, these results demonstrate that while Tfh cells specifically benefit from signals derived from the immunizing antigen, Tfr and Treg cells do not.

**Tfr and Tfh have different *TRBV* CDR3 length distributions**. We next performed TCR usage analysis in C57BL/6 mice, bearing a $Foxp3^{gfp}$ reporter system, immunized with a model antigen (OVA-IFA). To minimize the contamination with pre-existent Tfh and Tfr cells from prior immune responses, we used footpad immunization with collection of popliteal LNs at the peak of the GC response. As shown in Fig. 1g,h, there are negligible numbers of Tfh and Tfr cells in this anatomic location before immunization.

Taking advantage of the $Foxp3^{gfp}$ reporter system, we sorted Tfh ($CD4^+CXCR5^+PD-1^+Foxp3^-$), Tfr ($CD4^+CXCR5^+PD-1^+Foxp3^+$), Treg ($CD4^+CXCR5^-PD-1^-Foxp3^+$) and Tconv ($CD4^+CXCR5^-PD-1^-Foxp3^-$) cells from the draining LNs (Supplementary Fig. 2c). The *TRBV* repertoire of the different populations was then analysed by CDR3 spectratyping/Immunoscope[26]. The *TRBV* CDR3 length profiles (or spectratypes) of naive polyclonal $CD4^+$ T cells resemble Gaussian distributions[26] (Supplementary Fig. 3). Therefore, by comparing the CDR3 length usage for each *TRBV* of the four sorted populations to naive $CD4^+$ T cells (used as control population), we can detect variations to this polyclonal distribution. Indeed, in the Tfh population there is an over-representation of specific CDR3 lengths (in red, Fig. 5a). There are also some over-representations for the Tfr cells but the majority of these, besides not being common to Tfh cells, are also present in Treg cells (Fig. 5a). In fact, detailed analysis of specific *TRBV* segments can identify clonal expansions among Tfh cells that are absent in the other T-cell subsets (Fig. 5b, arrows). Also, we calculated a perturbation score[27] for every *TRBV* segment between all samples and the Tfr group average. Hierarchical clustering and principal component analysis (PCA) were performed using the calculated perturbation scores to reveal the divergent *TRBV* usage of all cell populations compared to Tfr group average. On the heatmap, Tfh cells present the higher perturbation scores (more divergent *TRBV* CDR3 length distributions from Tfr cells) and are clustered separately, while we observe a proximity between Tfr and Treg cells (Fig. 5c). Moreover, the same relation between sample populations can be observed when we perform PCA: the first two principal components, which describe 77.3% of the samples variability, separate the Tfh from the remaining samples, while clustering Tfr, Treg and Tconv together (Fig. 5d). Analysis of variance and multiple comparison analysis with Holm–Bonferroni correction for multiple sampling established the significance of the differences observed between populations, and between Tfh samples and each sample from the other populations (Supplementary Table 1).

Taken together, our data show that within LNs draining the immunizing site, Tfh cells exhibit clear oligoclonal expansions of specific *TRBV* CDR3 lengths. The same pattern of *TRBV* CDR3 usage is not observed within the Tfr population that retain distributions of CDR3 lengths usage similar to Treg cells.

**The TCR repertoire of Tfr cells resembles that of Treg cells**. The spectratyping data demonstrate that although Tfr and Tfh cell numbers increase after immunization, they present different *TRBV* CDR3 length usage. To further verify the different TCR usage between both populations, we sequenced the *TRA* of 1D2β mice, which express a fixed TCRβ chain and variable TCRα chains. This 1D2β mouse line, of C57BL/6 background, was established using nuclear-transferred embryonic stem (NT-ES) cells that had been generated using peripheral $CD25^{high}CD4^+$ T cells as donor of nuclei. The productively rearranged *TRB* gene of one NT-ES cell line was successfully transmitted to germ-line and the resulting 1D2β mice were mated with $Foxp3^{hCD2}$ and $TRA^{-/-}$ mice to generate $Foxp3^{hCD2}.TRB1D2.TRA^{-/WT}$ mice. Thus, 1D2β mice *TRA* sequencing provides a complete insight to the repertoire of the analysed populations, compared to WT mice, as the corresponding *TRB* does not vary.

Although these mice have a restricted *TRB* repertoire, we confirmed that their $CD4^+$ T cells were able to recognize OVA and differentiate into Tfh cells following OVA immunization (Fig. 6a). To show that 1D2β Tfh cells that arise after OVA immunization can specifically recognize the antigen, we performed *in vitro* proliferation assays (Fig. 6b). 1D2β mice were immunized with OVA-IFA, and 11 days later Tfh ($CD4^+CXCR5^+PD-1^+Foxp3^-$) cells from three individual mice were sorted and cultured with DCs pulsed with OVA (Supplementary Fig. 2d). Tfh cells from all three mice were able to proliferate with OVA signals but not with the control antigen βLG (Fig. 6b). Thus, these mice, while not being able to recombine the *TRB*, still have a repertoire capable of specifically recognizing OVA.

Five T-cell populations were sorted for *TRA* repertoire analysis on day 11 following OVA-IFA immunization from draining LNs of individual mice: Tfr ($CD4^+CXCR5^+PD-1^+Foxp3^+$); Tfh ($CD4^+CXCR5^+PD-1^+Foxp3^-$); Treg ($CD4^+CXCR5^-PD-1^-$

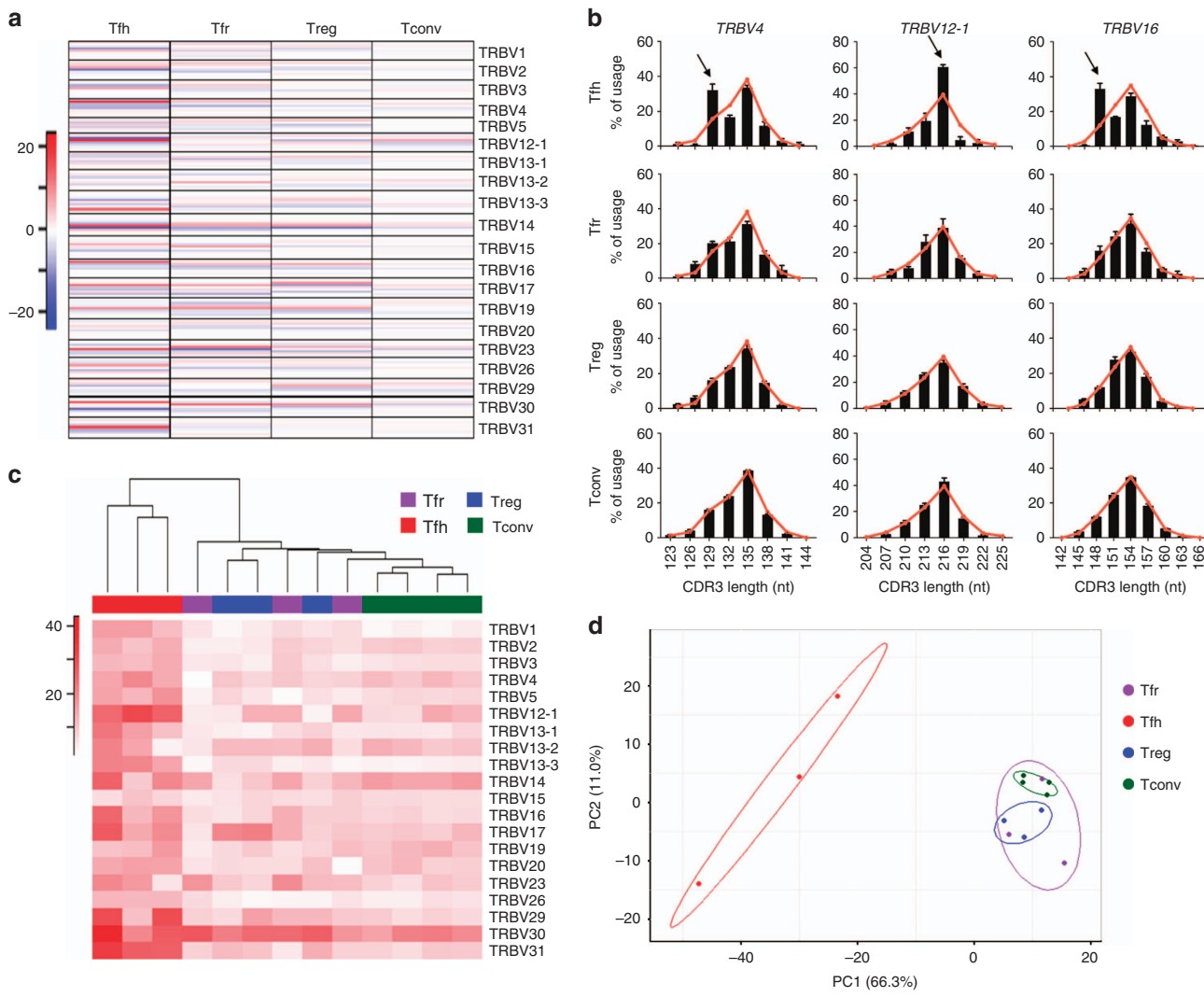

**Figure 5 | Tfh cells display antigen-induced oligoclonal proliferation that is absent from Tfr cells.** (**a**) Heatmap showing the differences between the percentage of usage for each CDR3 length of each *TRBV* of Tfh, Tfr, Treg and Tconv populations compared to the Gaussian-like distribution of CD4$^+$ naive T cells (used as a control population). Similar *TRBV* CDR3 length frequencies (compared to control population) are displayed in white, while higher/lower frequencies of specific CDR3 lengths are represented, respectively, in red and blue. The heatmap is representative of, at least, three independent experiments per population. (**b**) Distribution of CDR3-length usage for three representative *TRBV* segments where greatest variation is observed. Bars represent CDR3-length usage distribution for indicated populations, with the reference values (naive CD4$^+$ T cells) superimposed in red. Arrows indicate over-representation of a specific CDR3 length on Tfh cells, a putative consequence of clonal selection and expansion. Neither Treg nor Tfr cells show similar expansions (bar graphs below). Mean ± s.e.m. are represented on the bar graphs. (**c**) Hierarchical clustering of the samples from the four populations based on their *TRBV* perturbation scores, calculated using the Tfr group average as reference. Heatmap colour code indicates variations of *TRBV* scores between each sample and the average of Tfr group, while the dendrogram shows distance between sample populations. (**d**) Replicates from different T-cell subsets were projected according to the first two PCA components. Tfh samples are apart from all the other subsets. In all panels, for each T-cell population, we used at least three independent replicates, each one with cells sorted from 10–15 mice.

Foxp3$^+$); activated CD4$^+$ T (Tact: CD4$^+$CXCR5$^-$PD-1$^-$ Foxp3$^-$CD44$^+$); and non-activated CD4$^+$ T (Tconv: CD4$^+$ CXCR5$^-$PD-1$^-$Foxp3$^-$CD44$^-$) cells (Supplementary Fig. 2d). *TRA* gene was then specifically amplified and sequenced on an Illumina MiSeq platform. To perform an unbiased analysis, 9,000 TCR sequences were randomly selected from each data set per sample (9,000 being the lowest number of TCR sequences identified on a sample).

We started by verifying the number of common clonotypes between Tfr cells and other populations. The number of shared clonotypes is higher between Tfr and Treg cells than any other population (Fig. 6c and Supplementary Fig. 4a). However, we found that from the 9,000 TCR sequences the number of identifiable clonotypes was lower for Tfr, Tfh and Tact when

compared to Treg and Tconv. To determine whether this observation was in line with different clonality in the populations, we calculated a clonality score for each sample (Fig. 6d). Indeed, the clonality score was higher for the populations with lower numbers of identifiable clonotypes, as expected, given the fact that Tfh, Tact and Tfr have undergone cell proliferation, unlike Treg and Tconv. When we checked the frequency distribution of the 20 most frequent clonotypes for each sample across all samples, we observed that the 5 most abundant clonotypes represented in average up to ~50% of the total frequency for Tfh and Tact, and ~40% for Tfr cells (Fig. 6e). This was somehow expected since Tfr, Tfh and Tact almost do not exist before immunization and must expand on immunization, in contrast to Treg and Tconv (Fig. 6a). Remarkably, the most common

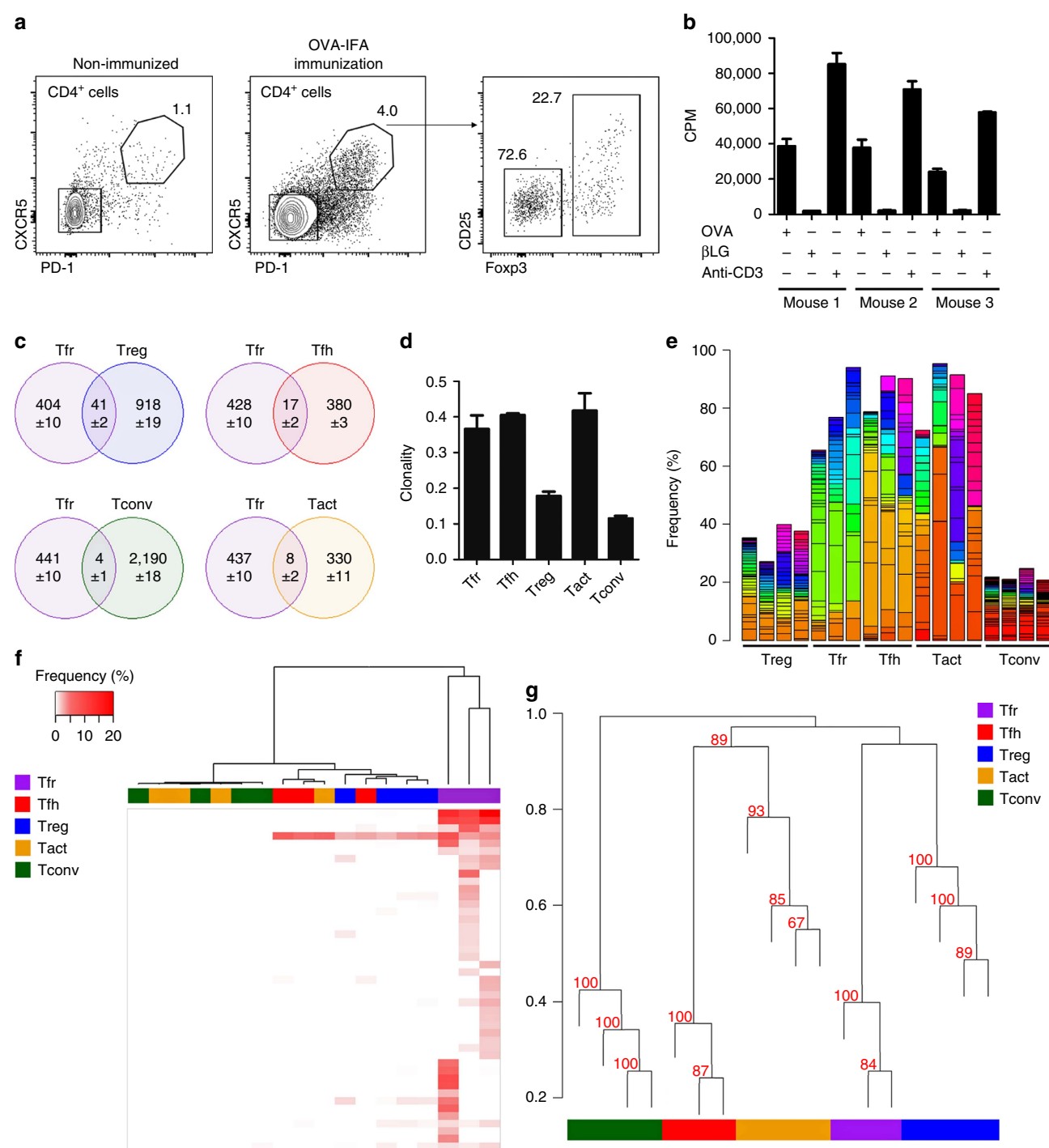

**Figure 6 | Deep sequencing analysis unveils a different repertoire between Tfr and Tfh cells.** (**a**) Frequency of follicular CD4$^+$CXCR5$^+$PD-1$^+$ T cells on popliteal LNs of 1D2β mice before (left), and 11 days after OVA-IFA immunization (middle), when both Tfr and Tfh cells are present (right). (**b**) Proliferation of Tfh cells from OVA-IFA-immunized 1D2β mice cultured with DCs pulsed with OVA or βLG (anti-CD3 was used as positive control). Cell proliferation was measured by 3H-thymidine incorporation. Mean ± s.e.m. of culture triplicates are presented and are representative of two independent experiments. (**c**) Venn diagrams showing the number of shared clonotypes between Tfr cells and the other four populations for mouse 1. Numbers are the mean ± s.d. of clonotypes identified after 100 iterations of the sampling process. Tfr cells have more common clonotypes with Treg cells than any other cell population. (**d**) Clonality score for the five populations. (**e**) Histogram of cumulative frequency across all samples of the union of the 20 most predominant clonotypes for each sample. Each colour corresponds to a unique clonotype. (**f**) Heatmap and hierarchical clustering of the 20 most predominant clonotypes of Tfr replicates across all samples. Tfr most predominant clonotypes are mostly shared with Treg samples, with the exception of one sequence that is common with Treg and Tfh samples. (**g**) Dendrogram showing the overall relation of all sequenced samples using Horn–Morisita index distance method. Bootstrap resampling was performed to calculate approximately unbiased (AU, in red) *P* values for each cluster. In the case of the Tfr and Treg cluster, AU = 96% thus the existence of the cluster is strongly supported by the data. For sequencing data, histograms present mean ± s.e.m. for *n* = 4, except for Tfh and Tfr where *n* = 3.

clonotypes from Tfh cells are shared in different mice (yellow), and the same happens for the most common Tfr clonotypes (green), although without a significant overlap between Tfh and Tfr clonotypes (Fig. 6e). To further investigate the proximity between Tfr and Tfh TCR repertoire, we investigated whether the 20 most predominant clonotypes from each Tfr sample are shared with other populations. We found that the cell population that shares more clonotypes with the predominant Tfr clonotypes is the Treg population (Fig. 6f). Among the most abundant clonotypes there is only one shared between all Tfr and Tfh samples, but that clonotype is also present on all Treg samples. The same approach was performed to obtain the 20 most predominant clonotypes for each Tfh sample (Supplementary Fig. 4b), where it is shown that Tfh cells present more shared clonotypes with Tact. Finally, we wanted to establish the closeness between all samples regardless of the observed expansions. For that we performed hierarchical clustering using Horn–Morisita index[28]. This index has been described as appropriate to compare immune repertoires since it is able to access the similarity between samples while taking into account the abundance of each clonotype in each sample. We found that Tfr and Treg samples are clustered together (approximately unbiased = 96%), indicating that Tfr samples have a repertoire closer to that of Treg cells, rather than to that of any of the other populations (Fig. 6g).

These results indicate that although Tfr cells undergo proliferation, their TCR is not specific for the antigen driving the GC reaction since the TCR usage has little in common with Tfh cells repertoire. Instead, Tfr cells present a TCR repertoire that closely resembles that of Treg cells.

## Discussion

During GC reaction the BCR undergoes affinity maturation leading to formation of higher-affinity receptors selected by Tfh cells. However, some mutations may lead to autoreactive receptors or receptors that are cross-reactive with autoantigens. Given the documented capacity of Tfr cells to prevent autoantibody-mediated autoimmunity[13–17], we investigated the possibility that the TCR usage by Tfh and Tfr populations from the same GCs is different: only the Tfh repertoire is biased towards the immunizing antigens.

We used three distinct approaches to address that issue: one approach based on cell transfer of T cells with defined TCR specificity; another approach using tetramers to identify antigen-specific Tfh and Tfr cells in WT mice; and a final approach based on the analysis of the TCR repertoire. All experiments led to similar conclusions, showing that T-cell clones specific for the immunizing antigen are enriched within the Tfh pool, but not within the Tfr cells. Our experiments do not directly show that Tfr cells recognize self antigens and, as a consequence, the antigen specificity of Tfr cells remains to be established. Nevertheless, as the TCR repertoire of Tfr cells is closer to that of Treg cells, rather than to that of any of the other populations, we speculate that Tfr cells share the same specificity as the Treg population, which is biased towards self antigens.

We cannot exclude that under certain conditions, Tfr cells may be specific for the immunizing antigen and differentiate from peripherally induced Treg cells. Indeed, a recent report claims that Tfr cells can differentiate from Foxp3− T cells and can be specific for the immunizing antigen[29]. However, even under those conditions tested, the percentage of tetramer-binding Tfr cells was only ~3% of total Tfr cells (which we did not observe on our experiments even after enrichment) and only a percentage of those Tfr cells presented markers indicating their differentiation from Foxp3− T cells. Moreover, these cells alone do not seem to substitute Tfr cells that originate from thymic Treg cells: Tfr cells seem to influence the affinity maturation process[30,31] and it has

been also shown that mice lacking Treg cells able to migrate into the GC have impaired affinity maturation[31].

We studied popliteal LNs in mice immunized in the footpad. In that location nearly all Tfh and Tfr cells are derived from GCs induced by the immunizing antigen (with negligible contaminants). As a consequence, the different TCR usage between Tfr and Tfh cells, as shown in Figs 5 and 6, suggests different antigen-specificity requirements. Furthermore, the fact that Tconv and Treg from draining LNs have Gaussian-like *TRBV* distributions and a low clonality score is in line with our observations with adoptively transferred TCR-transgenic cells: adoptive transfer of TCR-transgenic T cells into non-lymphopenic hosts (Figs 2c and 3d) leads only to a minor expansion of antigen-specific cells among Tconv (4–6%), unlike what is observed within Tfh cells (30–40%). In addition, the results obtained with tetramers corroborated our hypothesis: we could detect antigen-specific tetramer-binding Tfh, but not Tfr, cells indicating a different TCR usage.

The ontogenic origin of Tfh and Tfr cells has remained an important issue as it may be related to the distinct functional specialization of the two T follicular populations. Indeed, following immunization with an antigen in adjuvant, a process similar to what is observed following vaccination, Tfr cells associated with those induced GCs originate from pre-existing thymic Treg cells (Fig. 1). Such ontogenic proximity between Tfr and Treg has a repercussion on the TCR usage by the two populations: the TCR usage by Tfr cells remains largely similar to the TCR usage of Treg cells.

The different ontogeny of Tfr and Tfh cells suggests an attractive model for distinct functions of the two populations: while Tfh cells predominantly promote humoral responses targeting non-self antigens, Tfr cells prevent the generation of autoantibody-mediated autoimmunity. An expected consequence of this model is a different TCR repertoire of Tfh versus Tfr cells, which is confirmed by our present results. The ability of Tfr cells to regulate non-antigen-specific B-cell clones has been suggested[9]. The distinct range of antigen targets of Tfh and Tfr cells from the same GCs provides a molecular basis for such differential behaviour. It should be noted, however, that such distinction is not complete: it was shown that Tfr cells can partially regulate the amount of antibodies produced targeting a foreign antigen[8].

In conclusion, our data establish a different antigen specificity of Tfr and Tfh populations from the same GCs. Tfh cell repertoire comprises oligoclonal expansions in response to immunizing antigens. Such expansions are not observed in Tfr cells that bear a TCR repertoire resembling that of Treg cells, and thus, possibly, biased towards autoreactivity.

## Methods

**Mice and animal procedures.** C57BL/6, C57BL/6.Thy1.1, Foxp3$^{gfp}$, Foxp3$^{hCD2}$, Balb/c, TCRβ$^{-/-}$, DO11.10.Rag$^{-/-}$, OT-II.Rag$^{-/-}$, OT-II.Rag$^{+}$. Thy1.1.Thy1.2, P25 and 1D2β mice were bred and maintained in our institute-specific pathogen-free facilities. Animals of both sex (same sex per experiment) and with age ranged from 2 to 6 months were used. For animal studies, no randomization or blinding was done. Permission for animal experimentation was granted by ORBEA-iMM (the institutional Animal Welfare Body) and DGAV (Portuguese competent authority for animal protection). Animals were immunized subcutaneously in the footpad with different antigens: ovalbumin (Sigma-Aldrich, Catalogue#A5503); βLG (Sigma-Aldrich, Catalogue#L3908); OVA$_{323-339}$ (ISQAV-HAAHAEINEAGR) peptide; Ag85B$_{280-294}$ (FQDAYNAAGGHNAVF) peptide (Schafer-N); and with OVA$_{323-339}$ or Ag85B$_{280-294}$ conjugated with BSA (synthesized by Thermo Fisher Scientific or Schafer-N). Antigens were prepared by mixing the antigen solution 1:1 (v:v) with IFA or complete Freund's adjuvant, respectively (Sigma-Aldrich, Catalogue#F5506 and F5881) to a final concentration of 1.6 mg ml$^{-1}$ (proteins and peptides coupled to proteins) or 2 mg ml$^{-1}$ (peptides alone). A volume of 50 μl of emulsion was injected in each footpad. For adoptive cell transfers, purified CD4$^+$ T cells were injected intravenously in saline solution. In all experiments the immunization occurred 1 day after the adoptive cell transfer, and collection of popliteal LNs at day 11 following immunization.

**Flow cytometry and cell sorting.** For flow cytometry analysis and sorting, single-cell suspensions were obtained and stained with the following monoclonal antibodies: CD4 (RM4-5, dilution 1/200), CD19 (ebio1D3, dilution 1/100), CD25 (PC61.5, dilution 1/400), Foxp3 (FJK-16s, dilution 1/100), human CD2 (RPA-2.10, dilution 1/200), PD-1 (J43, dilution 1/100), Thy1.1 (HIS51, dilution 1/200), Thy1.2 (53-2,1, dilution 1/400), TCRβ (H57-597, dilution 1/100) and Vα2 (B20.1, dilution 1/200) from eBioscience; GITR (DTA.1, dilution 1/200), CXCR5 (2G8, dilution 1/50) and Vβ5.1, 5.2 (MR9-4, dilution 1/200) from BD Pharmingen; and CD4 (RM4-5, dilution 1/100) from Biolegend. Intracellular Foxp3 staining was performed using the Foxp3 Staining Set (eBioscience, Catalogue#00-5523-00) according to the manufacturer's instructions. OVA$_{323-339}$- or Ag85B$_{280-294}$-specific T cells were detected with a PE-(OVA)- or APC-(Ag85B)-conjugated MHC-II I-A$^b$ tetramer containing an OVA$_{329-337}$ (AAHAEINEA) or Ag85B$_{280-294}$ (FQDAYNAAGGHNAVF) peptide, respectively. Staining was performed for 1 h at room temperature[32]. Enrichment of tetramer$^+$ cells was performed using MACS cell separation system and anti-PE and anti-APC magnetic beads (Miltenyi Biotec, Catalogue#130-048-801 and 130-090-855). Samples were acquired on a BD LSR Fortessa flow cytometer. Acquisition data were analysed on FlowJo software (Tree Star). For cell sorting CD4$^+$ T cells for adoptive cell transfers were purified from spleen and mesenteric LNs using MACS cell separation system and anti-CD4 (L3T4) magnetic beads (Miltenyi Biotec, Catalogue#130-049-201). FACS, with monoclonal antibodies mentioned above, was performed on a BD FACSAria cell sorter. For flow cytometry analysis of cultured cells and tetramer-enriched cell samples, 10 μm latex counting beads were added to cell suspensions to obtain total cell counts (Counter Beckman).

**In vitro cultures.** Bone marrow-derived DCs were generated by culturing progenitors for 7 days in presence of granulocyte–macrophage colony-stimulating factor (PeproTech). Specific antigen loading was performed for 3 h at 37 °C in presence of 1 mg ml$^{-1}$ of protein. In OT-II Treg/WT Treg and Tfr/Tfh/Treg cell cultures, T cells were pre-incubated with Cell Trace Violet (Life Technologies) for tracking cell proliferation according to the manufacturer's instructions. A 3:2 ratio of CD4$^+$ T cells to DCs was used to a final number of $5 \times 10^4$ cells per well in the case of OT-II Treg/WT Treg cell cultures and $2.5 \times 10^4$ cells per well in the case of Tfr/Tfh/Treg cell cultures. Cells were co-cultured in the presence of 2 ng ml$^{-1}$ IL-2 (eBioscience, Catalogue#14-8021-64) and, in some conditions, 3 μg ml$^{-1}$ anti-CD3 (145-2C11, eBioscience) was added to the culture. After 3 days, cells were stained and analysed by flow cytometry. For Tfh proliferation assay, $2.0 \times 10^4$ Tfh cells were cultured with the same number of DCs. In the wells where unloaded DCs were cultured, 3 μg ml$^{-1}$ anti-CD3 was added to the culture. After 3 days of culture, cells were incubated with 1.0 μCi per well of 3H-thymidine (Perkin Elmer, Catalogue#NET027W001MC) at 37 °C for 6 h, collected on a Tomtec Harvester (Tomtec) and scintillation counted on a Microbeta Trilux (PerkinElmer).

**CDR3 length analysis.** RNA extraction from cell-sorted populations ($2 \times 10^5$–$5 \times 10^6$ cells) was performed using TRIzol (Life Technologies, Catalogue#15596). cDNA was amplified using Random Primers and SuperScript III Reverse Transcriptase (Invitrogen, Catalogue#48190011 and 18080-044). Both RNA extraction and cDNA synthesis were performed following the manufacturer's instructions. To perform CDR3 spectratyping[26], each obtained cDNA was divided into 23 parallel PCR reactions with a common Cβ reverse primer and 23 Vβ-specific forward primers (GoTaq DNA Polymerase from Promega, Catalogue#M7801, and primers from Life Technologies, Supplementary Table 2). Run-off reactions were done using dye-labelled Cβ primer. All primer sequences can be found on Supplementary Table 2. Run-off products were run on ABI 3130XL Automatic Sequencer (Applied Biosystems) together with GeneScan 500 ROX dye Size Standard (Applied Biosystems, Catalogue#401734) and consequently separated based on their nucleotide size. Gene Mapper software (Applied Biosystems) was used to obtain nucleotide length and area of each peak.

**Deep sequencing.** RNA from 1D2β mice sorted cell populations ($1.1 \times 10^4$–$1.0 \times 10^6$ cells) was extracted using RNeasy Mini kit (Qiagen, Catalogue#74104) following the manufacturer's instructions. Full-cDNA library was prepared using Mint-2 kit (evrogen, Catalogue#SK005), which introduces 5′-adapters to cDNA fragments, according to the manufacturer's instructions. TRA was then specifically amplified using Pfx DNA polymerase (Invitrogen, Catalogue#11708013) and a primer pair (Life Technologies) specific for the 5′-adapter and the C region of TRA gene. Primers used for TRA amplification can be found on Supplementary Table 2. Sequencing library was prepared using the Nextera kit (Catalogue#FC-121-1030 and FC-131-1002), in which each sample was barcoded, and sequenced using 250 bp paired-end illumina MiSeq technology (all illumina).

**CDR3 data analysis.** To have adequate representation of a complete TRBV repertoire, it is necessary to analyse at least $2 \times 10^5$ cells. To achieve this number of cells, it was necessary to pool draining LNs from 15 mice for each biological replicate. In our experiments, we used at least three biological replicates for each T-cell subset. To extract and analyse the data obtained for CDR3 fragment size we used ISEApeaks[33,34]. Briefly, this software quantifies the percentage of use of each

CDR3 length, obtained by dividing the area of CDR3 peaks by the total area of all peaks within the profile. On C57BL/6 mice, TRBV21 and TRBV24 are pseudogenes and thus ignored on the analysis. TRBV12-2 was also discarded since it could not be detected on three of the samples. To facilitate the comparison between samples and populations, a perturbation score[27] was computed to obtain the overall differences between TRBV CDR3 spectratypes of each sample and the average profiles of Tfr samples as control group. Calculated scores were used to perform the hierarchical clustering (using Euclidean distance and average linkage) and PCA.

**Deep sequencing data analysis.** Paired-end 250 bp illumina sequencing data were initially trimmed and subsequently merged using PEAR[35]. clonotypeR[36] toolkit was then used to perform TCR sequence annotation. Two samples had to be discarded due to low sequencing quality (mouse 3 Tfh sample and mouse 4 Tfr sample). Out of the 7,547,998 raw reads obtained for the 18 remaining samples, we identified 25,099 TCR clonotypes from 949,729 productive TCR sequences. For the samples to be comparable, the analyses were performed on 9,000 randomly selected TCR sequences for each data set as it was the lowest number of TCR sequences found in a data set. We repeated this sampling process 100 times to obtain the mean ± s.d. values presented on the Venn diagrams. The presented clonality metric is 1 − Pielou's evenness index[37], and can vary from 0 to 1 (more diverse to less diverse). The Pielou's evenness corresponds to the Shannon's entropy[38,39] (using log 2) for each sample divided by the number of unique clonotypes (in log 2) of the same sample. For the histogram of cumulative frequency, the 20 most predominant clonotypes were determined for each sample, and gathered across all samples to plot the cumulative frequency of those clonotypes for each sample. The 20 most predominant Tfr clonotypes were selected from each of the three Tfr samples and gathered into a list used to perform hierarchical clustering in all samples (using Euclidean distance and average linkage). The same was performed for the 20 most predominant clonotypes of each Tfh sample. Dendrogram of overall relation between all samples was obtained using Horn–Morisita index[28,40] as distance and average linkage. This index assesses the similarity between samples taking into account the abundance of each clonotype in the sample. Approximately unbiased P values were calculated for each cluster through 1,000 bootstrap resampling iterations[41].

**Statistical analysis.** Scatter plots and column bar graphs were obtained using GraphPad PRISM. Unless stated otherwise, n represents the number of individual mice analysed per experiment. To determine statistical significance, two-tailed non-parametric Mann–Whitney U-tests were performed, and $P < 0.05$ was deemed significant (in figures: *$P < 0.05$; **$P < 0.01$). A minimum of five mice per group were used on in vivo experiments to allow usage of non-parametric statistical tests. Clustering analysis, PCA, Venn diagrams, analysis of variance, pairwise multiple comparison analysis with Holm–Bonferroni correction for multiple sampling, statistical analysis and multivariate analysis were performed using R software (http://www.r-project.org/).

**Data availability.** Sequence data that support the findings of this study have been deposited in the Sequence Read Archive with the accession code SRP096953. All other relevant data are available from the corresponding author on reasonable request.

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

## Acknowledgements

We are grateful to Constantin Fesel, Margarida Correia-Neves and Bruno Cerqueira-Rodrigues for advice and reagents used in our studies. We acknowledge the NIH Tetramer Core Facility for providing MHC-II tetramers, and the Gene Expression Unit at Instituto Gulbenkian de Ciência where the deep sequencing was performed. The research was funded by Fundação para Ciência e Tecnologia (FCT) grants PTDC/SAU-IMU/120225/2010, HMSP-ICT/0034/2013 and FCT-FAPESP/19906/2014 (to L.G.). A.R.M. and S.C.P.A. are funded by FCT scholarships SFRH/BD/88030/2012 and SFRH/BDP/81391/2011, respectively. The work by D.K., A.S., E.M.-F., W.C. and F.J. has been funded by Assistance Publique-Hôpitaux de Paris, Université Pierre and Marie Curie (Paris VI), LabEx Transimmunom (ANR-11-IDEX-0004-02) and ERC Advanced Grant TRiPoD (322856). The work by J.F. has been supported by PIRSES-GA-2012-317893 (7th FP, EU) and BIOCAPS (FP7/REGPOT-2012-2013.1, EC) under grant agreement no. 316265.

## Author contributions

L.G. and J.F. conceived the idea and supervised the study; A.R.M., S.C.P.A., E.M.-F. and F.J. performed the experiments; A.R.M., S.C.P.A., E.M.-F., W.C., D.K., J.F. and L.G. analysed the data; A.S. and S.H. contributed with key reagents and software; A.R.M. and L.G. wrote the manuscript. All authors discussed the results and commented on the manuscript.

## Additional information

**Competing interests:** The authors declare no competing financial interests.

