## [Peer Review File · Nature Communications]

The original comments by the reviewers are in **bold**.

Our responses to the reviewers' comments are in blue.

Figures on this response are named R (e.g., *Figure R1*) to distinguish them from the main text figures (e.g., Fig. 1)

We would like to thank the reviewers for helpful suggestions that led to a much improved manuscript. In order to address the important points raised, we had to perform a large number of experiments including the sequencing of TRA of Tfr, Tfh, Treg, activated CD4⁺ T (Tact), and non-activated CD4⁺ T (Tconv) cells from individual 1D2 β mice (B6.Foxp3^{hCD2}.TRB^{1D2}.TRA^{-WT}). As suggested, each sorted population of each individual mouse was barcoded, allowing us a more sophisticated repertoire analysis. In addition, we improved the results with tetramers, now using two independent class-II tetramers. These new data led to a delay with the resubmission of the revised manuscript. We feel, however, that the current version is significantly improved.

We are now even more confident with the results as the independent experimental approaches validate our conclusion: the new sequencing data show that Tfr and Tfh cells have a different TCR repertoire, with the repertoire of Tfr cells being closer to Tregs. Adoptive transfer studies with TCR-transgenic cells support the view that T cells specific for the immunizing-antigen are preferentially recruited into the Tfh pool, but not into the Tfr pool. And finally, experiments with tetramers, as well as *in vitro* proliferation assays, established that wild-type Tfh and Tfr cells from immunized mice (bearing distinct TCR repertoires as shown by sequencing) have distinct TCR-specificity.

Overall the data support our conclusion that Tfh and Tfr cells from the same GCs have different TCR repertoires and while Tfh cells are specific for the immunizing antigen, Tfr cells are not.

Reviewers' comments:

Reviewer #1 (Remarks to the Author):

This manuscript by Graca and colleagues investigates the origins and TCR specificity of T follicular helper cells (Tfh) and T follicular regulatory cells (Tfr). While elegantly performed, the studies showing that Tfh cells originate from naïve CD4 T cells and Tfr cells originate from Treg are largely confirmatory. The TCR specificity of Tfh and Tfr cells is an important issue to address, but the studies performed require clarification and more in depth analyses.

1. Several aspects of Figure 1 need to be clarified. In the studies in Figure 1, the authors use congenic markers to track transferred T cells, but the identity of the Thy1.1 and Thy1.2 mice is not clear. The authors should move Fig 2a into Figure 1 so that the reader can appreciate that the OTII Rag+ mice have thymic FoxP3+ TCR transgenic cells. These data are important for understanding the transfer experiment with OTII-Rag+ cells in Fig 1c. In addition, the studies in Fig 1e are not clear. The text states that the mice are unimmunized but it appears that there is a comparison of naïve and OVA immunized mice. Please clarify.

In order to clarify the identity of Thy1.1 and Thy1.2, we changed the text and figure legend and included two diagrams (as Fig. 1b). We also moved the old panel of Fig. 2a into the new Fig. 1e, as suggested. Indeed, it facilitates the understanding of the data presented in Fig. 1. We apologize for our oversight in failing to mention the comparison between naïve and immunized mice. We changed the text, adding an explanation that the experiment included immunized and non-immunized mice to compare Tfr and Tfh numbers before and after immunization. We made changes to Fig. 1 and its legend.

2. How do TCR alpha and beta chain staining compare to the use of congenic markers to evaluate the TCR transgenic T cells in the studies in Figure 2? What is the overlap of TCR alpha and beta chain usage between Tfh and Tconv? What is the overlap of TCR alpha and beta chain usage between Treg and Tfr?

We thank the reviewer for this suggestion that we had not considered. Because in the cell transfers into WT mice there are very few events for the Treg/Tfr populations, we determined the Va2Vβ5 usage of OT-II.Rag⁺ cells in the cell transfer into TCRβKO experiment (*Figure R1* below). These data were added to the paper as Fig. 2f and demonstrate: “We found that, as previously described, a proportion of Treg cells (~30%) do not co-express the transgenic TCR chains Va2 and Vβ5 unlike Tconv cells which are virtually all double-positive. Nevertheless, upon adoptive transfer into TCRβKO mice and immunization, the percentage of OT-II.Rag⁺ Treg and Tfr cells co-expressing Va2 and Vβ5 remained unchanged (Fig. 2f). Therefore, we could not find a preferential enrichment of Va2 Vβ5 double-positive cells, more likely to be specific for the immunizing antigen, within the regulatory populations after OVA immunization.”

We believe these data further reinforce our results: if the transgenic TCR would provide an advantage for a cell to become Tfr, one would expect an enrichment of Va2 Vβ5 double-positive cells among the Tfr population of OVA-immunized mice. We found that there was no significant change. New experimental data added as Fig. 2f.

Figure R1 – Percentage of Va2 Vβ5 double positive cells on OT-II.Rag⁺ cell populations after transfer into TCRβKO mice. Included in the paper as Fig. 2f.

3. In Figure 2, what percentage of the transferred Treg become Tfr cells? Flow cytometry plots as well as graphs would be helpful to understand the percentage of Tfr cells that are antigen-specific. The use of MHC-II tetramers for OVA peptide would be informative.

We performed a new analysis on the cell transfers into TCRβKO mice to obtain these data. We found that, independently of the immunization, around 8% of the OT-II.Rag⁺ Treg cells differentiate into Tfr cells, while a higher percentage of OT-II Tconv differentiates into Tfh on mice immunized with OVA compared to βLG (*Figure R2*). These new experimental data were included in the manuscript as Supplementary Fig. 1a,b.

Figure R2 - OT-II Rag⁺ Treg cells differentiate into Tfr cells independently of the immunization (a) Gating strategy to determine the percentage of OT-II Treg and Tconv cells that differentiate into Tfr and Tfh cells, respectively. (b) While the percentage of OT-II Tconv that differentiates into Tfh is higher in mice immunized with OVA than βLG (left), the same percentage of OT-II Treg originates Tfr cells in both immunizations (right). Included in the paper as Supplementary Fig. 1a,b.

4. Further studies are needed to evaluate the antigen-specificity of Tfr cells. Similar to the studies with Treg in Fig 2g, the authors should conduct functional studies to evaluate Tfr responses to antigen *in vitro*. The authors should immunize polyclonal mice, purify Tfr cells from draining lymph node, and compare responses to the immunogen vs. control antigen. It would be informative to purify Tfr and Treg and compare their responses (expansion/cytokine production) to immunogen vs. control antigen *in vitro*.

We thank the reviewer for this suggestion. We feel the results of this experiment have improved significantly our manuscript. These experiments are difficult to perform given the fragility of follicular cells in culture that do not receive proliferative signals. We immunized mice with OVA-IFA, sorted defined cell populations, and performed *in vitro* cultures as suggested (Figure R3). We found that *in vitro* OVA-stimulation did not provide a significant advantage to Tfr cells when compared to β LG-stimulation or cultures with unloaded DCs (although some low background proliferation is present, as it generally happens with Treg cells stimulated *in vitro* in presence of IL-2). Tfr and Treg cultures were also performed. As expected, Tfr expanded and survived with OVA signals (positive control), while Treg cells did not (negative control). This experiment was added to the paper as Fig. 4f-h.

Figure R3 - Histograms of proliferation and bar graphs of total cell numbers of sorted Tfr ($CD4^+CXCR5^{high}PD-1^{high}Foxp3^+$), Tfh ($CD4^+CXCR5^{high}PD-1^{high}hCD2^-$), and Treg ($CD4^+CXCR5^+PD-1^+hCD2^+$) cells from OVA-immunized $Foxp3^{hCD2}$ reporter mice and cultured for three days in the presence of IL-2 with DCs loaded with OVA or β LG proteins or unloaded DCs with or without anti-CD3. Included in the paper as Fig. 4f-h.

5. In Figure 3a (refers to Fig. 4), the authors should add percentages to the flow cytometry plots and clarify if these are tetramer positive cells. Scatter plots rather than histograms would be helpful in interpreting the data in Figure 3A and a histogram showing tetramer positive CD44⁺ cells would be informative.

We thank the reviewer (and reviewer #2) to bring to our attention the lack of clarity of the tetramer experiments. Since this is a key piece of evidence, we generated new data in order to provide more compelling evidence.

We did not add percentages to the plots because this would not be very informative since an enrichment of PE-positive cells had been performed. Such enrichment is considered the standard method for staining with class II tetramers (Moon et al., 2009) (cited on the manuscript). For this new version of the manuscript, we performed new experiments with two different MHC-II tetramers and again performed enrichments for PE- and APC-positive cells. Therefore, we present event count instead of percentage on Fig. 4b,d. Nevertheless, a version of Fig. 4a-e with percentage values is presented here (Figure R4). The conclusions are the same, but we feel it is more correct to provide the data with the cell counts. The data are now presented as scatter plots instead of histograms, as suggested by the reviewer, and indeed that representation makes the data more compelling. Regarding the CD44 staining, we added this marker to the antibody panel on the new experiments. However, we do not think this marker could provide additional information as virtually all T follicular cells are CD44⁺ (Figure R5). As a consequence, we did not include these data. The new experimental data with tetramers were included in Fig. 4.

Figure R4 – Version of Fig. 4a-e from the manuscript presenting percentages on dot plots instead of counting values.

Figure R5 – CD44 expression on follicular CD4 T cells. Most of the follicular CD4 T cells (CXCR5⁺PD-1⁺, in red) express CD44, while this marker is only found on a minor percentage of the CXCR5⁺PD-1⁻ cells (in blue).

6. In Figure 4b (refers to Fig. 3), what percentage of the transferred Treg become Tfr cells. TCR alpha beta stains as well as Thy1.1/thy1.2 stains should be presented.

As we mentioned in point 3 (above), the number of events of Treg/Tfr OT-II on this experimental conditions is low, not allowing a correct analysis of the percentage of OT-II Tregs that differentiate into Tfr cells. We performed the suggested experiment by transferring the TCR-transgenic cells into TCRβKO mice, where a greater number of Treg/Tfr OT-II can be obtained. The results are represented in Figure R2 (under point 3) and were included as Supplementary Fig. 1a, b.

The reviewer was right in suggesting the display of the Thy1.1/Thy1.2 stains. Plots showing gating strategy that include Thy1.1/Thy1.2 staining were now added to Fig. 3.

7. The data that analyze TCR usage by polyclonal T cells following immunization are the most innovative aspect of the manuscript. However, the data should be analyzed in greater depth. What is the percentage of unique sequences and repeated sequences for each of the 4 populations (Tfh, Tconv, Tfr, Treg)? A clonality score is needed for each of the 4 populations. A Venn diagram could be used to show sequences that are shared between Treg and Tfr cells. How do precursor frequencies of TCRs in Treg and naïve T cells in unimmunized mice compare to TCRs present in Treg, Tfr, Tfh and Tconv after immunization? Is there selective expansion and clonality of Tfr cells?

We thank the reviewer for these suggestions. The analysis proposed could not be obtained from the spectratyping data because (1) cells were sorted from pools of 15 mice and (2) of the bias introduced by the parallel PCRs for each TRBV (from the spectratyping/immunoscope protocol). But given the importance of this information we decided to sequence the TRA of Tfr, Tfh, Treg, activated CD4⁺ T (Tact), and non-activated CD4⁺ T (Tconv) cells from individual 1D2β mice (B6.Foxp3^{hCD2}.TRB^{1D2}.TRA^{-WT}). These mice, besides being Foxp3 reporter (Foxp3^{hCD2}), have the TCRβ fixed and only one TCRα gene available for recombination, which allows the identification of the complete repertoire by sequencing the TRA alone.

A new results section has been introduced in the manuscript ("Tfr repertoire of TCRβ restricted mice is similar to Treg cells and different from Tfh cells."), as well as a new Fig. 6 and Supplementary Fig. 3. With the new data, it became possible to perform the analysis in greater depth, in individual mice, including the suggested studies (Venn diagrams, clonality score, evaluation of expansion and clonality of Tfr cells). We, therefore, decided to remove the previous sequencing data that did not contribute additional information.

8. The data in manuscript suggest that Tfr cells are specific for self-antigen but this is not shown. Recent work of Mark Jenkins (Nelson et al., Immunity 2015) identified multiple tetramers specific for self antigens in the naïve repertoire of C57BL/6 mice. These tetramers could provide a means to compare self -reactive TCR usage by Tregs in unimmunized mice with Treg and Tfr in immunized mice.

In the referred paper, Jenkins and colleagues have indeed performed an extensive work on populations specific for several peptides, using corresponding tetramers, to demonstrate that cross-reactivity between similar foreign and self peptides influences naïve cell population size. However, of the 22 tetramers used on the study, only two tetramers contained self antigens: the 2W tetramer on Act-2W transgenic mice (that express 2W as a self-peptide under the control of the actin promoter), and the MOG tetramer that was the only true self-antigen tetramer. Taking this into account, and with the knowledge that there are not many tetramers with self antigens available, we think that this approach is, at least for now, very difficult to accomplish.

Reviewer #2 (Remarks to the Author):

The study by Maceiras et al. addresses the TCR-specificity of Tfh and Tfr. To this, three different experimental models in the context of immunization were chosen: adoptive transfer of TCR-transgenic CD4 T cells, Tetramer-technology to identify antigen-specific Tfh and Tfr cells, and T cell receptor repertoire analysis of clonal families based on their CDR3 length.

The manuscript addresses an important question in the field: the antigen-specificity of Tfr cells and its implications on GC recruitment. Unfortunately, the data shown are still premature and do not add convincingly to this particular topic. The narrative and the rationale for experimentation is difficult to follow and the individual models used lack detailed explanation and justification. Lastly, the methodology and analytical strategies for TCR repertoire analyses were described without the necessary detail and are difficult to interpret. Collectively, the current state of the manuscript is found to be not appropriate for publication in Nature Communications.

Major comments:

Differences in clonality between Tfh and Tfr, as well as Tfr originating from Foxp3 positive cells rather than Tfh have been described before. The authors hypothesize that the polyclonal composition of the TCR repertoire of Tfr compared to a more oligoclonal repertoire of Tfh most likely leads to a different recruitment pattern to the draining LN, potentially linked to the different function of these CD4 T cell subsets during an ongoing immune response. This hypothesis is very attractive but unfortunately no data, such as functional properties of these subsets or more detailed clonal analysis of dominant clones that were exclusively found within Tfh or shared between subsets are provided to support this conclusion. Also the described presence of antigen-specific Tfr in the draining LN was not further discussed and rather interpreted as negligible background. The authors ignore a potential impact of the given adjuvant or the nature of the provided antigen on the differentiation of Tfr from other cells than FOXP3+ cells (Linterman and colleagues, Nature Communications, Jan 2016). Therefore, the data generated in transgenic mouse models and tetramer technology are not sufficient to support the hypothesis.

To our better knowledge, the clonality of Tfr cells has not been described before. Therefore, our paper is the first study addressing this problem.

The reviewer is right in identifying the lack of data regarding the analysis of the dominant clones as a shortcoming of our initial manuscript. We investigated this issue and included a new figure (Fig. 6) addressing this point (more details under specific points below). Overall the new data confirm our conclusions.

Regarding the percentage of antigen-specific Tfr found in the draining LNs, we concluded that it was not antigen-dependent and consequently negligible because the same percentage of antigen (OVA)-specific Tfr cells was observed independently of the immunizing antigen (OVA vs β LG).

As for Linterman and colleagues results, mentioned by the reviewer, they were published while our manuscript was being reviewed and we were not aware of them. However, we believe, based on our experience with IFA and CFA, that the reported impact of the adjuvant on Tfr induction may have an explanation not considered in that report.

Indeed, from the new tetramer experiments that we performed to complete Figure 4 on our manuscript, we verified that the same ratio of Tfr:Tfh is obtained using IFA or CFA as adjuvant (*Figure R6a* of this document). The effect reported by Linterman and colleagues may be a consequence of the use of peptides alone (in that paper the authors use MOG-peptides in IFA – an adjuvant without proteins) versus the presence of proteins that leads to more physiological germinal centres (as the other group relies on immunization with MOG peptide in CFA, that has plenty of proteins). Furthermore, when we transferred CD4 OT-II.Rag^{-/-}, we almost could not see conversion of those cells into Foxp3⁺ CD4 cells, which indicates that the IFA environment does not induce significant numbers of pTregs, although a small amount of pTreg cells can be identified (See *Figure R6b,c*).

Figure R6 – IFA does not preferentially induce conversion into pTregs. (a) Tfr:Tfh ratios are maintained independently of the adjuvant used (IFA or CFA). Ratio of Tfr:Tfh cells within follicular T cell (CD4⁺CXCR5⁺PD-1⁺ cells). (b) DO11.10.Rag^{-/-} and OT-II.Rag^{-/-} CD4 T cells, when transferred into Balb/c and C57BL/6 hosts, respectively, followed by immunization with OVA in IFA, almost do not differentiate into pTreg cells. (c) Percentage of DO11.10.Rag^{-/-} and OT-II.Rag^{-/-} that become Foxp3⁺. For b and c, data were extracted from experiments presented on Fig. 1a-d of the manuscript. Mean±SEM for n=5 in both graphs.

To confirm that the protein content, and not the nature of the adjuvant, was playing a significant role on the magnitude of the GC reaction (as the ratio of Tfr:Tfh was constant as shown in Figure R6a), we investigated the impact of immunization with two distinct peptides, alone or combined with a carrier protein (BSA), in IFA. We found that, while we can induce GC B cells in all immunizations with IFA as adjuvant, the percentage of GC B cells is higher when peptides are coupled to a protein (see Figure R7). Possibly B cells do not easily recognize and mount responses to small peptides alone as, although B cells can recognize linear epitopes, not all peptides can trigger good antibody responses as previously shown (Agarwal et al., 1998). In some cases, peptides need to be introduced attached to other structures in order to be recognized as antigen (such as carrier proteins). These data were not included in the manuscript.

Figure R7 – While IFA is able to support the induction of GC responses, higher B cell responses are induced when a protein is used as antigen compared to a peptide alone. (a) Plots showing percentage of GC B cells after immunization with OVA₃₂₃₋₃₃₉ in IFA or OVA₃₂₃₋₃₃₉-BSA in IFA. (b) Percentage of GC B cells, determined as CD19⁺GL-7⁺Fas⁺ on mice immunized OVA₃₂₃₋₃₃₉ or OVA₃₂₃₋₃₃₉-BSA in IFA. (c-d) Same as in (a-b) but regarding immunizations with Ag85b₂₈₈₋₂₉₄ or Ag85b₂₈₈₋₂₉₄-BSA in IFA. All plots are representative of five mice for each immunization. On bar graphs Mean±SEM is presented for n=5.

Using spectratyping, the authors attempt to detail the T cell repertoire present in immunization-induced Tfr and Tfh. Unfortunately, the chosen technique lacks the level of sophistication needed for conclusive statements and fails to identify the whole body of Tfh and Tfr clonotypes. For subsequent analyses following NGS of CDR3 amplicons, the text and data does not inform whether analyses were done on the clonal (individual CDR3 sequences) or TCRBV-BJ combination (families) level. As such, statements about cellular ontogeny or true clonal dominance cannot be made. Also, claims about statistical significance of different population compositions were made but not shown. The authors argue that the naïve T cell CDR3 lengths are normally distributed, but fail to provide evidence. Further statistical testing for deviation from such distribution in Tfh is missing. They further state to derive a metric of 'difference' per TCRBV but no detail is provided, and lastly, neither rationale for, nor interpretation of principal component analysis of the data is detailed and provided. Collectively, the reported TCR repertoire data is missing the required details that would allow for sufficient interpretation.

We agree with the reviewer that spectratyping on its own does not allow a conclusive statement. This concern was the reason for combining different experimental approaches to independently tackle the same issue. We believe that the combined approach allows us now to reach a conclusive statement, in particular given the extended information we now provide with tetramer stainings, and also with the new data with TCR sequencing from individual mice. Taken together, it is conclusive that the Tfr and Tfh TCR repertoires from the same germinal centres are different as stated in the title.

We reviewed our manuscript and included new experimental data that followed important suggestions, in order to improve our results. We also discarded the old Fig. 6, replacing it with more compelling data (new Fig. 6).

We have now added a Supplementary Table 1 with the results of statistical tests performed, as well as a Supplementary Fig. 2 data showing the normal-like distribution of naïve T cell CDR3 lengths in line with what was previously published (Pannetier et al., 1993). We show in Fig. 5a the differences between each sample CDR3 length distributions toward naïve CD4⁺ T cell. Coloured lines indicate perturbed CDR3 lengths for each TRBV in comparison with naïve CD4 T cells. Moreover, on Fig. 5b, as we explain in the manuscript, we provide a detailed analysis of 3 TRBV segments for which Tfh cells have increased frequencies on specific CDR3 lengths compared to the reference values. These results further illustrate the existence of clonal expansions among Tfh cells that are absent in the other T cell subsets.

The metric of “difference” per TRBV, referred as perturbation score and used to perform the clustering and PCA, was previously described and published in the given reference (Gorochov et al., 1998). This score corresponds to the overall deviation between the CDR3 length distribution of a specific TRBV of a sample and the CDR3 distribution for the same TRBV of a chosen reference population.

The principal component analysis (PCA) is a dimensionality reduction technique used to emphasize variability and identify strong patterns in large datasets. As the name indicates, it calculates in an unsupervised fashion the components that describe the variability between samples in a way that the first principal component accounts for as much of the variability in the data as possible, while each succeeding component describes as much as possible of the remaining variability. We present a PCA that represents 77.3% of the total variability between samples perturbation scores and clearly demonstrates that Tfh cells are very different from the remaining samples in terms of CDR3 length distributions using Tfr cells as reference. These data were now confirmed with TCR sequencing from individual mice (Fig. 6).

Finally, we sequenced the TRA of Tfr, Tfh, Treg, activated CD4⁺ T (Tact), and non-activated CD4⁺ T (Tconv) cells from individual 1D2 β mice (B6.Foxp3^{hCD2}.TRB^{1D2}.TRA^{-WT}).

We strongly believe the new data presented on a new results section, on Fig.6, and on Supplementary Fig.3 significantly improved the manuscript.

Specific comments to individual Figures:

Figure 1: The main text states that the peak GC response is at day 11. No kinetics to establish this time point are shown nor a relevant reference is provided. Tfr are believed to be established within the first 48h of priming: More details to prove the optimal time point used would be helpful to validate the model and the statement.

We thank the reviewer for pointing out the need to clarify the choice of the time point. In fact, our previous work included the kinetics of accumulation of Tfh and Tfr cells in a primary GC response (Wollenberg et al., 2011). Our data were more recently confirmed by a different group (see Figure 2 in Vanderleyden et al., 2014). These references were added to the manuscript.

Our studies do not directly address the time for Tfr priming, we simply take advantage of the time point when both Tfr and Tfh cells can be isolated in greater numbers (Vanderleyden et al., 2014; Wollenberg et al., 2011).

Each genetic background leads to a distinct amount of Foxp3+ Tfr cells (Balb/c: 16.7%, B16: 9% and TCRbeta -/-: 6.1%). How does this discrepancy in induced Tfh reflect a "normal proportion", as stated in the main text, if only one mouse per strain and experiment is shown? Statistics of several mice per group and adoptive transfer experiment would be helpful to establish a normal proportion.

Fig 1d-e: Which mouse strain was used for these figures and what is the number of mice shown?

We thank the reviewer for identifying the need for a better clarification regarding the use of two different strains of mice. The reason for using different genetic backgrounds is indeed to account for the anticipated biological heterogeneity. Our data show that, in spite of the expected quantitative differences between the strains, the same qualitative outcome is present. As the experiment was performed on several mice from each strain, we obtained graphs showing that those differences are due to mouse heterogeneity, but the overall Tfr:Tfh ratio is the same in both strains (see Figure R8), although slightly lower than when mice do not receive TCR-transgenic cells (compare with Figure R6a). The experiment where only CD4 OT-II.Rag⁺ were transferred into TCRβKO mice is not comparable as these mice only received TCR-transgenic antigen-specific CD4 T cells.

The mouse strain used on the old Fig. 1d,e (now Fig. 1g,h) was C57BL/6 and we are presenting the results of 3 mice. This information was clarified in the text and in the figure legend (see Figure R9).

Figure R8 – Tfr:Tfh ratios are similar in Balb/c and C57BL/6 backgrounds. Ratios of Tfr:Tfh cells within follicular T cell (CD4+CXCR5+PD-1+ cells) from Balb/c or C57BL/6 mice that received DO11.10.Rag⁺ or OT-II.Rag⁺ cells were immunized with OVA in IFA one day later. Analysis was performed 11 days after immunization (data were extracted from experiments presented on Fig. 1a-d). Mean±SEM for n=5.

Figure R9 – New version of Fig. 1d,e (Fig. 1g,h on the new version). (g) Relative frequency of T follicular cells in popliteal LNs from non-immunized C57BL/6 mice. (h) Absolute number of Tfh and Tfr cells within popliteal LN from non-immunized C57BL/6 mice compared to OVA-immunized mice. Mean \pm SEM is presented for n=3.

Figure 2: In four mice ~3% Tfr and ~28%Tfh cells differentiated and were found in the draining LN following immunization, in these experiments collected at day 12. Why not day 11?

In all experiments, the LNs collection was performed at day 11 after immunization. However, in this experiment we started counting the time on the day the cells were transferred (day 0), being the immunization on the following day. We changed the text, stating cell transfer happened at day -1, and immunization on day 0, in order to keep it coherent throughout the manuscript, always having collections of LNs on day 11, thus avoiding this confusion.

Legend of Figure 2 d and f is confusing. Are OT-IIs or cells defined by congenic marker shown within the target population?

The presented percentage refers to the percentage of OT-II cells (defined by congenic marker Thy1.2) within each of the four populations. We improved the legend.

What is the expected and average ratio of Tfh to Tfr in the context of these specific immunization strategies?

As the reviewer can observe on Figure R10, the average ratio of Tfr:Tfh is, as previously observed (see Figure R6a and Figure R8 for comparison), slightly lower on mice that received OT-II.Rag⁺ CD4 cells and were immunized with OVA compared to the ones that were immunized with β LG. This effect is probably due to the presence of a non-physiological high number of the antigen-specific cells (TCR-transgenic cells) upon immunization.

Could a distinct adjuvant recruit more Tfr cells?

We performed experiments directly comparing IFA and CFA and, as we have stated before, the same ratio of Tfr:Tfh cell is observed in both IFA and CFA immunization (shown in Figure R6a). Other immunizations, however, should be tested as they may have specific effects on the GC reaction.

Figure R10 – Ratios of Tfr:Tfh cell after immunization with OVA or β LG in IFA in C57BL/6 mice that received OT-II.Rag⁺ CD4 cells.

What is the threshold for recruited antigen-specific Tfr to control an ongoing GC reaction in these experiments? And are the recruited ~3% Tfr truly insufficient to regulate the GC reaction in these in these experiments? Can one truly conclude a lack of recruitment of antigen-specific Tfr to the GC using these models and experiments or is a distinct mechanism responsible for a consistent ratio between induced Tfh and Tfr in these mice? Fig e-f shows increased fractions of cells, but the overall ratio between Tfh and Tfr seems to be similar to panel d.

Although on Fig. 2c (Fig. 2d on the previous version) only 28% of the Tfh are OT-II cells, we believe that most of the remaining 72% are also OVA-specific Tfh cells, albeit derived from endogenous cells and therefore not identified with the congenic marker. Within the Tfr pool >95% of Tfr are not derived from TCR-transgenic cells. Moreover, the ratio of OVA-specific Tfh:Tfr cells is not the same in all immunizations: the same number of TCR-transgenic cells observed within the Tfr population, in TCR β KO mice immunized with OVA, is also present following immunization with the control antigen (β LG), where the number of recruited transgenic-TCR cells into Tfh pool is lower (20% in both immunizations in the case of Tfr and 50% vs 15% for OVA vs β LG immunizations in the case of Tfh). These results suggest that the observed recruitment of TCR-transgenic cells as Tfr cells is not to maintain Tfh:Tfr ratios of antigen-specific cells but instead at the level of non-specific effects.

Also, all subsequent experiments (tetramers, spectratyping, TCR sequencing) suggest that Tfr and Tfh cells from the same GCs have different TCRs.

Fig 2 g-h suggest that OT-II Tregs can proliferate in response to their antigen. Confusingly there is also a 67% proliferation shown in g to DCs-betaLG and 42.7% to DCs alone. This is a high background and not reflected in the summary (N=2 seems not sufficient to account for this variability) of the data shown in h, statistics on this N is not acceptable.

The legend was misleading in suggesting N=2. In fact, the graph represents triplicates from an experiment that was performed twice (N=3 x 2). However, the homogeneity of the results was quite striking.

The background Treg proliferation observed without specific TCR signalling when Tregs are stimulated with DCs in presence of IL-2 has been previously described by (Zou et al., 2010), and also observed on the Supplementary Fig. 4d of (Levine et al., 2014). We performed a new experiment that shows the dependence on IL-2 on the background proliferation of Treg cells (*Figure R11*). Indeed, when we repeated the experiment using 2 ng/ml of IL-2, instead of the 5 ng/ml initially used, we obtained the same results, but with lower background proliferation and total cell numbers at the end of culture (we substituted the results obtained with 5 ng/ml for these new ones) (*Figure R12*). In any case, the conclusion regarding specific stimulation in presence of OVA is undisputed.

Figure R11 – Treg unspecific proliferation is dependent on exogenous IL-2. Treg cells from WT mice were cultured with DCs with or without anti-CD3 and in the presence of different concentrations of IL-2. Total number of cells in the end of culture per well (left) and Percentage of dividing cells defined by CTV MFI in the end of culture (right).

Figure R12 – WT and OT-II.Rag⁺ Treg cells were cultured in presence of bone marrow DCs loaded with OVA or β-LG and 5 ng/ml of IL-2 (a, b) or 2 ng/ml (c, d). Cultures in the presence of unloaded DCs with or without anti-CD3 were used as positive and negative control, respectively. (a, c) Histograms are representative of Treg cell proliferation at the end of the culture. (b, d) Quantification of the number of proliferating cells. (c) and (d) were included in the paper, as Fig. 2g,h, in substitution of the old experiment performed with 5 ng/ml of IL-2.

Figure 4: The induction of tetramer-specific Tfh/Tfr cells following immunization. This is an elegant approach to recapitulate the above obtained data in a distinct model/experimental system. However the data provided are not convincing since CFA alone should not induce OVA-specific cells. Among the "large pool of antigens provided" within CFA, as stated in the main text, the model antigen OVA is not present. Thus the data are difficult to interpret.

We thank the reviewer for considering this approach elegant. We believe, however, that the experimental design was not properly explained and, as a consequence, misled the reviewer: we were comparing mice immunized with a cocktail of proteins (CFA immunization) with another group that received the same cocktail of proteins (CFA) in addition to OVA₃₂₃₋₃₃₉ peptide (OVA₃₂₃₋₃₃₉-CFA immunization). In the first case (CFA immunization), we did not expect to obtain Tfh nor Tfr specific for OVA (because OVA was absent), but we were expecting large numbers of Tfh and Tfr cells induced following the immunization with many proteins present in CFA (negative control). As for the OVA₃₂₃₋₃₃₉-CFA immunization, as expected, we detected OVA-specific Tfh cells (tetramer⁺), but not Tfr cells. Here the same proteins of CFA were present (and many Tfh/Tfr are likely induced to those proteins), but OVA is also present leading to induction of OVA-specific Tfh cells, but not Tfr cells. For this new version of the paper, we present a new experiment where we detected OVA tetramer⁺ cells and Ag85b tetramer⁺ cells on mice immunized with 4 different antigens in combination with 2 adjuvants: (OVA₃₂₃₋₃₃₉ in IFA, OVA₃₂₃₋₃₃₉ in CFA, OVA₃₂₃₋₃₃₉ coupled to BSA in IFA, Ag85b₂₈₀₋₂₉₄ in IFA, Ag85b₂₈₀₋₂₉₄ in CFA, and Ag85b₂₈₀₋₂₉₄ coupled to BSA in IFA). We also improved the explanation of the experimental design on the new version of the paper.

The summary of the data (4b) attempts to reflect the finding shown as histogram (4a). The numbers used in 4b seem to be different and also based on a high background staining (CFA alone average of ~130 in b) but only 24 counts in a) than shown in a.

The numbers presented in 4a and 4b were different because in 4a we presented the count of positive cells in the flow cytometer analysis, while in 4b we presented the total tetramer⁺ cells in the draining LNs (calculated from the FACS counts). Nevertheless, we clarified those issues in the legend, text, and material and methods as we present the results of the new data in a similar way. Please, see also reply to reviewer #1, point 5.

Why was CFA used as adjuvant now?

We used CFA, and not IFA, because we needed a negative control immunized with proteins without OVA peptide in order to compare to the same proteins with OVA peptide (the only case where tetramer⁺ cells should be expected). However, and as stated before, we performed new experiments using more antigens and adjuvants for better analysis and comparisons.

Figure 5: The colormap in (a) shows deviation from naïve repertoires not only for Tfh, but also for Tfr, Treg.

It has been described that, although Tregs have a highly diverse TCR repertoire, their repertoire is different from naïve Tconv cells (Hsieh et al., 2004; Hsieh et al., 2006). Thus, the deviation of Treg cells from the naïve CD4 repertoire was expected. Since Tfr cells mainly originate from Treg cells, although some proliferation does occur, we were anticipating that Tfr spectratyping would be different from Tconv cells. The new experiments with TCR sequencing (Fig. 6) confirm that the TCR repertoire of Tfr is more related to Tregs than to any other cell subset.

(b) only shows a selected collection of TCRBV families, overlaying the reference distribution would be helpful.

We display the information in several panels: the first provides the difference between each cell population and the reference distribution (Fig. 5a); the second, an example of selected TRBVs that illustrate some of the most significant changes (Fig. 5b); the third, an overall quantification of all the deviations between all samples and the reference distribution (Fig. 5c). Nevertheless, as suggested by the reviewer, we included in panel 5b the superimposed reference distributions for naïve CD4⁺ T cells for those three TRBVs, and as supplementary data the reference distribution for all TRBVs.

The metric for (c) is not accessible to the reader.

The metrics used for clustering analysis are stated on the correspondent analysis section of the methods. Regarding the perturbation score used to perform the clustering, and as referred before, it is described in the reference provided in the main text and in the CDR3 data analysis section (Gorotchov, 1998). These are standard methods used for spectratyping analysis. We rewrote some text to clarify the method.

(d) No statement about appropriateness for PCA is given, why are Tfh so different, even in an inbred model? What do the first two components reflect?

The principal component analysis (PCA) is a technique used to emphasize variation and bring out strong patterns in large datasets. The presented PCA is composed by two components that describe 77.3% of the total variability between samples. It is an unsupervised method and the variability described by the principal components is not pre-selected. Therefore, this method allows an easier visualization of samples' differences. As it is a standard method for datasets analysis, our description was brief. We have now revised the text to provide more information. Regarding the difference of the Tfh cells from other samples it is probably a consequence of their unique clonal expansion. As better observable in the new sequencing data, the differences between Tfh samples even in inbred models may be due to the fact that, although Tfh cells have some clonotypes that are shared by the three samples and account for a large proportion of the population, there are several other clonotypes, with a lower frequency in the population, that are not shared (Fig. 6).

Figure 6: If (b) is a top 1% fraction of (a), why are the patterns of (b) not represented in (a); with the color code (0~4%) equal in both, why is less purple in (a)? The hierarchical clustering dendrogram for the clones is unnecessary.

The patterns were not represented because the data presented were different: while in (a) we presented the TRBVJ usage, in (b) we presented the Tfh 1% most predominant clonotypes regardless of TRBVBJ usage. Regarding the colour code, as the reviewer may observe, the scale in (a) would go up to 5 while the scale in (b) would only go up to 3, indicating that 5% frequencies could be observed in (a) while only 3% frequencies could be observed in (b). Nevertheless, these data were removed from the manuscript. We believe that the new data on TCR sequencing are more powerful in allowing the study of individual mice, and the previous data would become redundant (Fig. 6). We also believe that the new data are displayed in a more understandable way.

Reviewer #3 (Remarks to the Author):

The Authors describe in their manuscript "T follicular helper (Tfh) and T follicular regulatory (Tfr) cells have different TCR-specificity" the different response of the two cell types to an antigen in mice. They show among others due tetramer experiments that Tfh cells respond to a OVA peptide with an expansion which they not observed for the Tfr cells. Further the author use two different technologies to investigate the TCR repertoire of different cell types.

I have some concerns especially with the last point.

1. It is not clear why the authors use two different methods to address the same question. Spectratyping and NGS sequencing. Which additional results can be obtained due the spectratyping which can not be obtained from the sequencing alone? It is even written in the paper that the results in the CDR3 length overlap between this methods.

We performed the spectratyping analysis as a first approach to determine whether we would observe differences in CDR3 lengths usage between Tfr and Tfh cells. Since we observed a rather global modification of the repertoire, we performed sequencing on all the PCR products from the spectratyping in order to obtain a more detailed information regarding the TCR clonotype usage of these populations (the overlapping of the two methods was only mentioned as a validation of both methods/results). Nevertheless, and in order to improve the manuscript's quality, we decided to eliminate the previous sequencing data as it became redundant with the new data (described below).

2. Following point one: The same PCR product was used for Spectratyping and NGS sequencing which are expected to produce the same results. A separate library creation optimized for sequencing approaches with barcoding every single mice would be a better validation for the spectratyping results as well for the following CDR3 comparison.

We have performed a new sequencing experiment where we sequenced the TRA of Tfr, Tfh, Treg, activated CD4⁺ T (Tact), and non-activated CD4⁺ T (Tconv) cells from individual 1D2 β mice (B6.Foxp3^{hCD2}.TRB^{1D2}.TRA^{WT}). As suggested, each sorted population of each individual mouse was barcoded. The analysis was performed on the usage of individual clonotypes for each sample. However, we cannot compare these sequencing results with the spectratyping data as the number of analysed cells is much lower (individual mice vs pool of 15 mice). In any case, the new experimental data are in line with the conclusion that the TCR repertoire of Tfr and Tfh cells from the same GCs is different. We provide the new experimental data as Fig. 6.

3. My major concern is that the sequencing results only show that the Tfh cells are clonally expanded whereas Tfr are not. Tfh present a limit number of cells, which were clonally expanded. The other groups show a more even distribution of their repertoire due to the absence of clonal expansion. This is expected taken into account that the Tfh expansion were shown in the previous flow cytometer results. The Tfr cells show no expansion and therefore a different more evenly distributed repertoire than the Tfh cells. If I understood it right the author indicate that based on the difference of the repertoire that the Tfr cell don't respond due to their different TCR repertoire. Which is not supported by the data. The Tfr show the expected similarity to the not expanded cell population, not presenting a different subpopulation. The observed difference of the Tfh is based on their expansion and consequently reduced diversity, which lead to the observed difference in CDR3 length compared to the Tfr cells. In my opinion it is not valid to compare this two groups with each other in the provided way. At least it should be normalized for the clonal expansion by comparing just the actually observed CDR3 as amino acid sequence and not only the CDR3 length after clonal expansion.

In light of the new data obtained following suggestions by the reviewers, we now know that Tfr cells, as expected from a population that proliferates upon immunization, present some level of expansion (see clonality score in Fig. 6d). However, as can be observed on Fig. 6f and Supplementary Fig. 3b (also on *Figure R13a,b*), the clonotypes with higher frequency on Tfr and Tfh samples are shared within samples of the same population, but not between Tfr and Tfh populations. Moreover, even though Tfr cells present a clonality score close to Tfh cells, when we perform hierarchical clustering using Horn-Morisita index as distance method, we verify that the more closely related population to Tfr cells is indeed Treg cells (Fig. 6g or *Figure R13c*). In conclusion, we believe that the new sequencing data demonstrate that Tfr cells are not specific for the

immunizing antigen (more frequent clonotypes are not shared with Tfh cells), while having a repertoire closer to Treg cells.

We also want to stress that the reason for using different experimental approaches is to independently validate our conclusion. Therefore, the tetramer data (Fig. 4), the proliferation of T cell subsets sorted from immunized mice following *in vitro* stimulation (a great suggestion from reviewer #1) (Fig. 4), as well as the adoptive cell transfer experiments (Fig. 2 and 3), all reinforce the same conclusion.

Figure R13 – Differences between Tfr and Tfh TRA repertoires. (a) Heatmap and hierarchical clustering of the 20 most abundant clonotypes of Tfr replicates across all samples. (b) Same as in (a) but regarding Tfh replicates. (c) Dendrogram of the overall relation between samples using Horn-Morisita index distance method.

4. Correspondingly the claimed similarity between Tfr and Treg could be completely explained by clonal expansion regardless of TCR similarity. The high clonal expanded low diversity Tfh on one side and on the other side the not expanded high diversity Tconv cells leads consequently to a higher similarity of the other two cell subsets.

In light of the new sequencing data, the similarity of Tfr and Treg cells cannot be explained by the lack clonal expansion regardless of TCR similarity, since Tfr cells present some expanded clones (that are however distinct from the ones present in Tfh cells). Moreover, the Horn-Morisita index method used, that takes into account the shared clonotypes and their abundance within the compared samples, shows that the clones of Tfr cells are more related to Tregs, even though the two populations have different clonality scores.

5. Of interest would be to compare the results within the Tfh group as well. Because the samples are sequenced a comparison of the obtained CDR3 regions in more detail would be possible.

Although it is not the objective of this study to evaluate differences between Tfh samples, it can be observed on Supplementary Fig. 3b (Figure R13b), and to some extent in Fig. 6e, that Tfh cells have some clonotypes that account for a large proportion of the population that are shared by the three samples (i.e., the same clonotypes are present in three different mice). However, there are several other clonotypes, with a lower frequency in the population, that are not shared.

Further would be the analyzing the individual mice response of interest and not in groups of 15. This could lead to more detailed results about how similar the TCR response between the mice's really is.

This is a key suggestion. The pooling of mice was a technical issue due to the number of cells necessary for the spectratyping analysis and limited number of cells even upon immunization. However, we felt our manuscript would be significantly improved with the addition of data from individual mice. Therefore, we analysed the TCR repertoire of 5 populations from individual mice as mentioned above. These new data are presented in the new Fig. 6. Data from the old Fig. 6 became redundant and were deleted.

6. Method part: I would like to know the oligo sequences used for V gene amplification as well as the used reverse primer as well some more data about the obtained sequencing depth as well of the different samples.

We added the list of primers as Supplementary Table 2. These primers are also listed in (Pannetier et al, PNAS 1993).

In summary, the major claim of this manuscript that Tfh and Tfr cells have a different TCR specificity, which is the reason for their different function, is not supported through the provided data. Its only show that Tfh cells clonally expands after immunization whereas Tfr cells are not.

We believe that our manuscript is now significantly improved with the input of several suggestions from the reviewers. One reason to use different experimental approaches was to obtain independent evidence that, overall, would allow us to reach a conclusion supported by the evidence. We especially believe that the new sequencing data support our hypothesis that Tfr and Tfh cells have different TCR repertoire. The tetramer staining and *in vitro* proliferation assays establish that the different TCR usage is linked with distinct TCR-specificity. Finally, the data on cell transfers of TCR-transgenic cells provide evidence that cells specific for the immunizing-antigen are preferentially recruited into the Tfh pool, but not into the Tfr pool.

Overall our data support our conclusion that Tfh and Tfr cells from the same GCs have different TCR repertoires and while Tfh cells are specific for the immunizing antigen, Tfr cells are not.

REFERENCES

- Agarwal, A., Nayak, B.P., and Rao, K.V. (1998). B cell responses to a peptide epitope. VII. Antigen-dependent modulation of the germinal center reaction. *J Immunol* *161*, 5832-5841.
- Gorochov, G., Neumann, A.U., Kereveur, A., Parizot, C., Li, T., Katlama, C., Karmochkine, M., Raguin, G., Autran, B., and Debre, P. (1998). Perturbation of CD4+ and CD8+ T-cell repertoires during progression to AIDS and regulation of the CD4+ repertoire during antiviral therapy. *Nat Med* *4*, 215-221.
- Hsieh, C.S., Liang, Y., Tyznik, A.J., Self, S.G., Liggitt, D., and Rudensky, A.Y. (2004). Recognition of the peripheral self by naturally arising CD25+ CD4+ T cell receptors. *Immunity* *21*, 267-277.
- Hsieh, C.S., Zheng, Y., Liang, Y., Fontenot, J.D., and Rudensky, A.Y. (2006). An intersection between the self-reactive regulatory and nonregulatory T cell receptor repertoires. *Nat Immunol* *7*, 401-410.
- Levine, A.G., Arvey, A., Jin, W., and Rudensky, A.Y. (2014). Continuous requirement for the TCR in regulatory T cell function. *Nat Immunol* *15*, 1070-1078.
- Moon, J.J., Chu, H.H., Hataye, J., Pagan, A.J., Pepper, M., McLachlan, J.B., Zell, T., and Jenkins, M.K. (2009). Tracking epitope-specific T cells. *Nat Protoc* *4*, 565-581.
- Pannetier, C., Cochet, M., Darche, S., Casrouge, A., Zoller, M., and Kourilsky, P. (1993). The sizes of the CDR3 hypervariable regions of the murine T-cell receptor beta chains vary as a function of the recombined germ-line segments. *Proc Natl Acad Sci U S A* *90*, 4319-4323.
- Vanderleyden, I., Linterman, M.A., and Smith, K.G. (2014). Regulatory T cells and control of the germinal centre response. *Arthritis Res Ther* *16*, 471.
- Wollenberg, I., Agua-Doce, A., Hernandez, A., Almeida, C., Oliveira, V.G., Faro, J., and Graca, L. (2011). Regulation of the germinal center reaction by Foxp3+ follicular regulatory T cells. *J Immunol* *187*, 4553-4560.
- Zou, T., Caton, A.J., Koretzky, G.A., and Kambayashi, T. (2010). Dendritic cells induce regulatory T cell proliferation through antigen-dependent and -independent interactions. *J Immunol* *185*, 2790-2799.